# ANATOMICAL STRUCTURE-AWARE IMAGE DIFFERENCE GRAPH LEARNING FOR DIFFERENCE-AWARE MEDICAL VISUAL QUESTION ANSWERING

## ABSTRACT

To contribute to automating the medical vision-language model, we propose a novel Chest-Xray Different Visual Question Answering (VQA) task. Given a pair of main and reference images, this task attempts to answer several questions on both diseases and, more importantly, the differences between them. This is consistent with the radiologist's diagnosis practice that compares the current image with the reference before concluding the report. For this task, we propose a new dataset, namely MIMIC-Diff-VQA, including 698,739 QA pairs on 109,790 pairs of images. Meanwhile, we also propose a novel expert knowledge-aware graph representation learning model to address this problem. We leveraged expert knowledge such as anatomical structure prior, semantic and spatial knowledge to construct a multi-relationship graph to represent the image differences between two images for the image difference VQA task. Our dataset and code will be released upon publication. We believe this work would further push forward the medical vision language model.

## 1 INTRODUCTION

Several recent works focus on extracting text-mined labels from clinical notes and using them to train deep learning models for medical image analysis with several datasets: MIMIC (Johnson et al., 2019), NIH14 (Wang et al., 2017) and Chexpert (Irvin et al., 2019). During this arduous journal on vision-language (VL) modality, the community either mines per-image common disease label (Fig.1. (b)) through Natural Language Processing (NLP), or endeavors on report generation (Fig.1. (c) generated from (Nguyen et al., 2021)) or even answer certain pre-defined questions (Fig.1. (d)). Despite significant progress achieved on these tasks, the heterogeneity, systemic biases and subjective nature of the report still pose many technical challenges. For example, the automatically mined labels from reports in Fig.1. (a) is obviously problematic because the rule-based approach that was not carefully designed did not process all uncertainties and negations well (Johnson et al., 2019). Training an automatic radiology report generation system to directly match the report appears to avoid the inevitable bias in the common NLP-mined thoracic pathology labels. However, radiologists tend to write more obvious impressions with abstract logic. For example, as shown in Fig.1. (a), a radiology report excludes many diseases (either commonly diagnosed or intended by the physicians) using negation expressions, e.g., no, free of, without, *etc.* However, the artificial report generator could hardly guess which disease is excluded by radiologists.

Instead of thoroughly generating all of the descriptions, VQA is more plausible as it only answers the specific question. As shown in Fig. 1, the question could be raised exactly for "is there any pneumothorax in the image?" in the report while the answer is no doubt "No". However, the questions in the existing VQA dataset ImageCLEF (Abacha et al., 2019) concentrate on very few general ones, such as "is there something wrong in the image? what is the primary abnormality in this image?", lacking the specificity for the heterogeneity and subjective texture. It often decays VQA into classification. While VQA-RAD (Lau et al., 2018) has more heterogeneous questions covering 11 question types, its 315 images dataset is relatively too small.

To bridge the aforementioned gap in the visual language model, we propose a novel medical image difference VQA task which is more consistent with radiologists' practice. When radiologists make

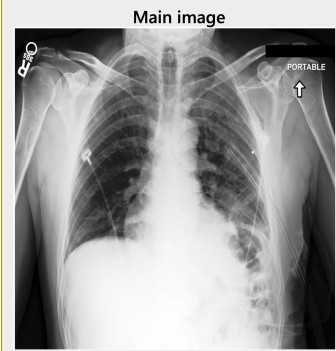

**Main image**

**(a) Ground Truth Report:**

No change in the positioning of the left-sided chest tube. Re demonstration of fractures of the fifth and sixth left posterior ribs. A pleural line is not present on the current study. Residual left lung atelectasis is present. The right lung is clear. Heart size is normal. No focal consolidation. No pleural line is detected to indicate residual left pneumothorax . Left lower lobe atelectasis persists, in the setting of possible splinting given the posterior rib fractures .

**(b) Text Mined Label from MIMIC**

Atelectasis
Fracture
**Pneumothorax (incorrect)**

**(c) Generated Report**

compared with the prior study , there is a new left apical pneumothorax with a small left apical **pneumothorax**. the left chest tube is in unchanged position . there is persistent elevation of the left hemidiaphragm and left basilar **atelectasis** . the right lung is clear . there is a small left **pleural effusion (unrelated diseases)** . heart is normal in size .

**(d) ImageCLEF VQA-MED questions**

is there something wrong in the image?
what is the primary abnormality in this image?

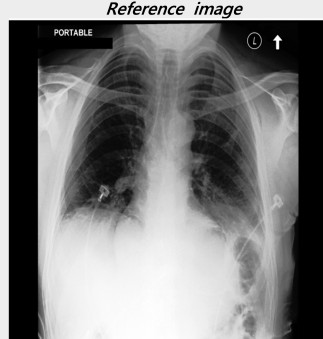

**Reference image**

**(e) Ground Truth Report**

Comparison to ___, 07:47. Removal of the left chest tube. No evidence of pneumothorax. Mild basal atelectasis. Mild cardiomegaly. No pneumonia.

**(f) Our medical difference VQA questions**

What abnormality is seen in this image?
Atelectasis, Fracture.
Is there pneumothorax in this image? No
Where is the fracture? Posterior ribs
Where is the atelectasis? Left lower lobe
What is the level of heart size? Normal
which view is this image taken? AP view
What has changed compared to the reference image?
The main image has an additional finding of fracture. The main image is missing the finding of cardiomegaly.

Figure 1: (a) The ground truth report corresponding to the main(present) image. The red text represents labels incorrectly classified by either text mining or generated reports, while the red box marks the misclassified labels. The green box marks the correctly classified ones. The underlined text is correctly generated in the generated report. (b) The label "Pneumothorax" is incorrectly classified because there is NO evidence of pneumothorax from the chest x-ray. (c) "There is a new left apical pneumothorax" → This sentence is wrong because the evidence of pneumothorax was mostly improved after treatment. However, the vascular shadow in the left pulmonary apex is not very obvious, so it is understandable why it is misidentified as pneumothorax in the left pulmonary apex. "there is a small left pleural effusion" → It is hard for a doctor to tell if the left pleural effusion is present or not. (d) The ImageCLEF VQA-MED questions are designed too simple. (e) The reference(past) image and clinical report. (f) Our medical difference VQA questions are designed to guide the model to focus on and localize important regions.

diagnoses, they compare current and previous images of the same patients to check the disease's progress. Actual clinical practice follows a patient treatment process (assessment - diagnosis - intervention - evaluation) as shown in Figure2. A baseline medical image is used as an assessment tool to diagnose a clinical problem, usually followed by therapeutic intervention. Then, another follow-up medical image is retaken to evaluate the effectiveness of the intervention in comparison with the past baseline. In this framework, every medical image has its purpose of clarifying the doctor's clinical hypothesis depending on the unique clinical course (e.g., whether the pneumothorax is mitigated after therapeutic intervention). However, existing methods can not provide a straightforward answer to the clinical hypothesis since they do not compare the past and present images. Therefore, we present a chest x-ray image difference VQA dataset, MIMIC-Diff-VQA, to fulfill the need of the medical image difference task. Moreover, we propose a system that can respond directly to the information the doctor wants by comparing the current medical image (main) to a past visit medical image (reference). This allows us to build a diagnostic support system that realizes the inherently interactive nature of radiology reports in clinical practice.

MIMIC-Diff-VQA contains pairs of "main"(present) and "reference"(past) images from the same patient's radiology images at different times from MIMIC(Johnson et al., 2019) (a large-scale public database of chest radiographs with 227,835 studies, each with a unique report and images). The question and answer pairs are extracted from the MIMIC report for "main" and "reference" images with rule-based techniques. Similar to (Abacha et al., 2019; Lau et al., 2018; He et al., 2020), we

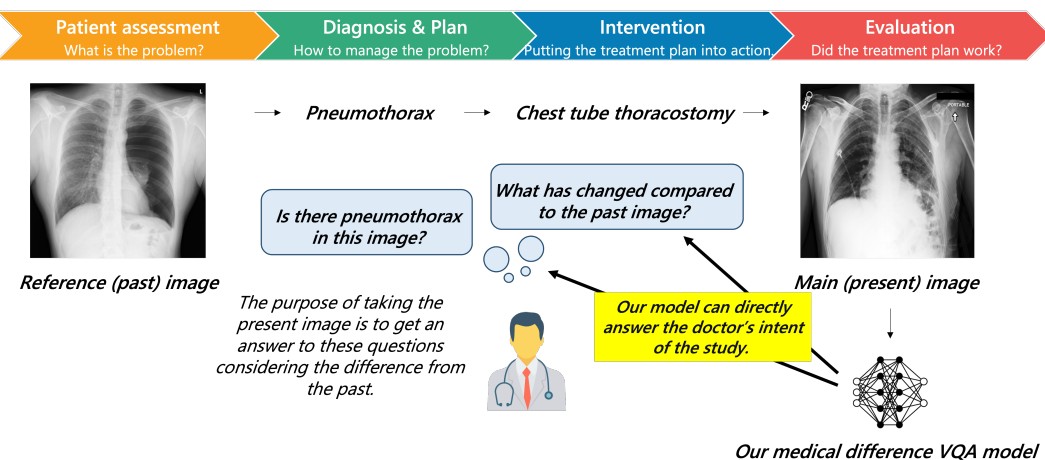

Figure 2: Clinical motivation for Image difference VQA.

first collect sets of abnormality names and attributes. Then we extract the abnormality in the images and their corresponding attributes using regular expressions. Finally, we compare the abnormalities contained in the two images and ask questions based on the collected information. We designed seven types of questions:1. abnormality, 2. presence, 3. view, 4. location, 5. type, 6. level, and 7. difference. In our MIMIC-Diff-VQA dataset, 698,739 QA pairs are extracted from 109,790 image pairs. Particularly, *difference* questions answer pairs inquiry on the clinic progress and change on the "main" image compared to the "reference" image as shown in Fig. 1(e).

The current mainstream state-of-the-art image difference method only applies to synthetic images with small view variations,(Jhamtani & Berg-Kirkpatrick, 2018; Park et al., 2019) as shown in Fig. 3. However, real medical image difference comparing is a very challenging task. Even the images from the same patient show large variances in the orientation, scale, range, view, and nonrigid deformation, which are often more significant than the subtle differences caused by diseases as shown in Fig. 3. Since the radiologists examine the anatomical structure to find the progression of diseases, similarly, we propose an expert knowledge-aware image difference graph representation learning model as shown in Fig. 3. We extract the features from different anatomical structures (for example, left lower lung, and right upper lung) as nodes in the graph.

Moreover, we construct three different relationships in the graph to encode expert knowledge: 1) Spatial relationship based on the spatial distance between different anatomical regions, such as "left lower lung", "right costophrenic angle", etc. We construct this graph based on the fact that radiologists prefer to determine the abnormalities based on particular anatomical structures. For example, "Minimal blunting of the left costophrenic angle also suggests a tiny left pleural effusion."; 2) Semantic relationship based on the disease and anatomical structure relationship from knowledge graph (Zhang et al., 2020). We construct this graph because of the fact that diseases from the same or nearby regions could affect each other's existence. For example, "the effusions remain moderate and still cause substantial bilateral areas of basilar atelectasis."; 3) Implicit relationship to model potential implicit relationship beside 1) and 2). The graph feature representation for each image is learned as a weighted summation of the graph feature from these three different relationships. The image-difference graph feature representation is constructed by simply subtracting the main image graph feature and the reference image graph feature. This graph difference feature is fed into LSTM networks with attention modules for answer generation(Toutanova et al., 2003).

**Our contributions are summarized as:**

1)We collect the medical imaging difference question answering problem and construct the first large-scale medical image difference question answering dataset, MIMIC-Diff-VQA. This dataset comprises 109,790 image pairs, containing 698,739 question-answering pairs related to various attributes, including abnormality, presence, location, level, type, view, and difference.

Figure 3: Anatomical structure-aware image-difference graph for medical image difference visual question answering.

2) We propose an anatomical structure-aware image-difference model to extract the image-difference feature relevant to disease progression and interventions. We extracted features from anatomical structures and compared the changes in each anatomical structure to reduce the image differences caused by body pose, view, and nonrigid deformations of organs.

3) We develop a multi-relationship image-difference graph feature representation learning method to leverage the spatial relationship and semantic relationship ( extracted from expert knowledge graph) to compute image-difference graph feature representation, generate answers and interpret how the answer is generated on different image regions.

## 2 METHODS

**MIMIC-Diff-VQA dataset.** We introduce our new MIMIC-Diff-VQA dataset for the medical imaging difference question-answering problem. The MIMIC-Diff-VQA dataset is constructed following an Extract-Check-Fix cycle to minimize errors. Please refer to Appendix. A.2 for the details on how the dataset is constructed. In MIMIC-Diff-VQA, each entry contains two different chest x-ray images from the same patient with a question-answer pair. Our question design is extended from VQA-RAD, but with an additional question type of "difference". In the end, the questions can be divided into seven types: 1) abnormality, 2) presence, 3) view, 4) location, 5) type, 6) level, and 7) difference. Tab. 1 shows examples of the different question types.

Table 1: Selected examples of the different question types. **See Table 5 in Appendix A.2 for the full list.**

| Question type | Example |
| --- | --- |
| Abnormality | what abnormality is seen in the left lung? |
| Presence | is there evidence of atelectasis in this image? |
| View | which view is this image taken? |
| Location | where in the image is the pleural effusion located? |
| Type | what type is the opacity? |
| Level | what level is the cardiomegaly? |
| Difference | what has changed compared to the reference image? |

The image pairs are selected from the MIMIC (Johnson et al., 2019) dataset, and each image in an image pair is from the same patient. A total of 109,790 image pairs are selected from MIMIC, and 698,739 questions are constructed. We also balance the "yes" and "no" answers to avoid possible

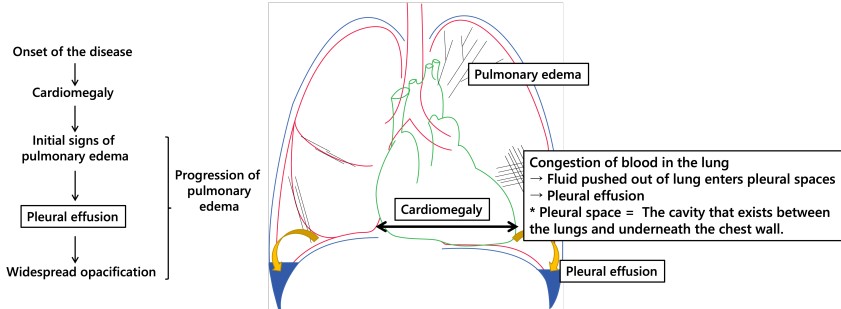

(a) Progression from cardiomegaly to edema and pleural effusion

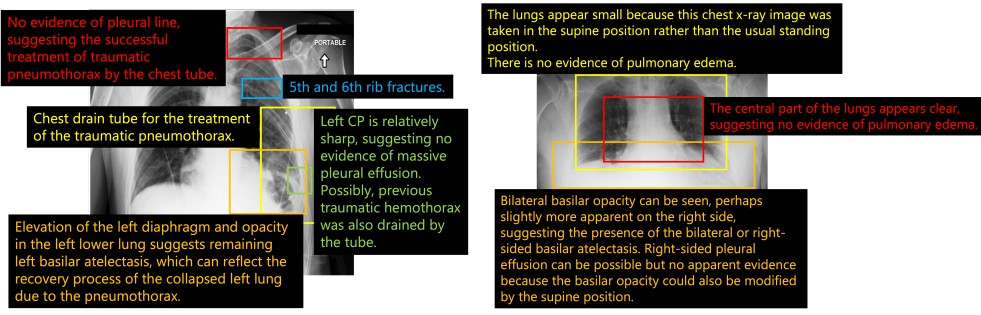

(b) Radiologist's annotation example 1          (c) Radiologist's annotation example 2

Figure 4: Illustration of the progression of diseases and two X-ray annotation examples.

bias. The statistics regarding each question type can be seen in Fig. 6 in Appendix. The ratio between training, validation, and testing set is 8:1:1.

**Problem Statement.** Given an image pair $(\mathbf{I}_m, \mathbf{I}_r)$, consisting of the main image $\mathbf{I}_m$ and the reference image $\mathbf{I}_r$, and a question $\mathbf{q}$, our goal is to obtain the answer $\mathbf{a}$ of the question $\mathbf{q}$ from image pair. In our design, the main and reference images are from the same patient.

**Expert Knowledge-Aware Graph Construction and Feature Learning.** As shown in the left Fig.3, previous work on image difference question answers in the general image domain. They create paired synthetic images with identical backgrounds and only move or remove the simple objects from the background. The feature of image difference was extracted by simply comparing the feature on the same image coordinates. Unfortunately, even the medical imaging of the same patients shows significant variations due to the pose and nonrigid deformation. The change of pose, scale, and range of the main image and reference image in Fig. 3 are strongly different compared to the disease change (pleural effusion changed from small to moderate). If we use the general image difference methods, the computed image differences related to the pose change will dominate, and the subtle disease changes will be neglected. To better capture the subtle disease changes and eliminate the pose, orientation, and scale changes, we propose to use an expert knowledge-aware image difference graph learning method by considering each anatomical structure as a node and comparing the image changes in each anatomical structure just as radiologists, which consist of the following parts:

**Anatomical Structure, Disease Region Detection, and Question Encoding.** We first extract the anatomical bounding boxes and their features $\mathbf{f}_a$ from the input images using pre-trained Faster-RCNN on the MIMIC dataset (Ren et al., 2015; Karargyris et al., 2020). Then, we train a Faster-RCNN on the VinDr dataset (Pham et al., 2021) to detect the diseases. Instead of directly detecting diseases on the given input images, we extract the features $\mathbf{f}_d$ from the same anatomical regions using the extracted anatomical bounding boxes. The questions and answers are processed the same way as (Li et al., 2019; Norcliffe-Brown et al., 2018). Each word is tokenized and embedded with Glove ((Pennington et al., 2014)) embeddings. Then we use a bidirectional RNN with GRU (Cho et al., 2014) and self-attention to generate the question embedding $\mathbf{q}$.

**Multi-Relationship Graph Module.** After extracting the disease and anatomical structure, we construct an anatomical structure-aware image representation graph for the main and reference image. The multi-relationship graph is defined as $\mathcal{G} = \{\mathbf{V}, \mathcal{E}_{sp}, \mathcal{E}_{se}, \mathcal{E}_{imp}\}$, where $\mathcal{E}_{sp}, \mathcal{E}_{se},$ and $\mathcal{E}_{imp}$ represent the edge sets of spatial graph, semantic graph and implicit graph, each vertex $\mathbf{v}_i \in \mathbf{V}, i = 1, \cdots, 2N$ can be either anatomical node $\mathbf{v}_k = [f_{a,k}\|\mathbf{q}] \in \mathbb{R}^{d_f + d_q}, f_{a,k} \in \mathbf{f}_a,$ for $k = 1, \ldots, N,$ or disease node $\mathbf{v}_k = [f_{d,k}\|\mathbf{q}] \in \mathbb{R}^{d_f + d_q}, f_{d,k} \in \mathbf{f}_d,$ for $k = 1, \ldots, N,$ representing anatomical structures or disease regions, respectively. Both of these two types of nodes are embedded with a question feature as shown in Fig. 3. $d_f$ is the dimension of the anatomical and disease features. $d_q$ is the dimension of the question embedding. $N$ represents the number of anatomical structures of one image. Because each disease feature is extracted from the same corresponding anatomical region, the total number of the vertex is $2N$.

We construct three types of relationships in the graph for each image: 1) **spatial relationship**: We construct spatial relationships according to the radiologist's practice of identifying abnormalities based on specific anatomical structures. For example, "the effusions remain moderate and still cause substantial bilateral areas of basilar atelectasis"; "Elevation of the left diaphragm and opacity in the left lower lung suggests remaining left basilar atelectasis" as shown in Fig. 4b; "The central part of the lungs appears clear, suggesting no evidence of pulmonary edema." as shown in Fig. 4c. In our MIMIC-Diff-VQA dataset, questions are designed for the spatial relationship, such as "where in the image is the pleural effusion located?" as shown in Tab. 1. Following previous work (Yao et al., 2018), we include 11 types of spatial relations between detected bounding boxes, such as "left lower lung", "right costophrenic angle", etc. The 11 spatial relations includes `inside` (class1), `cover` (class2), `overlap` (class3), and 8 other directional classes. Each class corresponds to a 45-degree of direction. We define the edge between node i and the node j as $a_{ij} = c$, where c is the class of the relationship, $c = 1, 2, \cdots, K$, K is the number of spatial relationship classes, which equals to 11. When $d_{ij} > t$, we set $\mathbf{a}_{ij} = 0$, where $d_{ij}$ is the euclidean distance between the center points of the bounding boxes corresponding to the node $i$ and node $j$, $t$ is the threshold. The threshold $t$ is defined to be $(l_x + l_y)/3$ by reasoning and imitating the data given by (Li et al., 2019).

2) **Semantic relationship**: The semantic relationship is based on two knowledge graphs, including an anatomical knowledge graph from (Zhang et al., 2020), as shown in Fig. 8a, and a label occurrence knowledge graph built by ourselves, as shown in Fig. 8b. If there is an edge linking two labels in the Knowledge graph, we connect the nodes having these two labels in our semantic relationship graph. The knowledge graph can include abstracted expert knowledge and depicts the relationships between diseases. These relationships play a crucial role in disease diagnosis. Multiple diseases could be interrelated to each other during the course of a specific disease. For example, in Fig. 4a, a progression from cardiomegaly to edema and pleural effusion is shown. Cardiomegaly, which refers to an enlarged heart, can start with a heart dysfunction that causes congestion of blood in the heart, eventually leading to the heart's enlargement. The congested blood would be pumped up into the veins of the lungs. As the pressure of the vessels in the lungs increases, fluid is pushed out of the lungs and enters pleural spaces causing the initial sign of pulmonary edema. Meanwhile, the fluid starts to build up between the layers of the pleura outside the lungs, i.e. pleural effusion. Pleural effusion can also cause compression atelectasis. As pulmonary edema continues to progress, widespread opacification in the lung can appear. These can be verified in actual diagnostic reports. For example, "the effusions remain moderate and still cause substantial bilateral areas of basilar atelectasis"; "Bilateral basilar opacity can be seen, suggesting the presence of the bilateral or right-sided basilar atelectasis" as shown in Fig. 4c.

3) **Implicit relationship**: a fully connected graph is applied to find the implicit relationships that are not defined by the other two graphs. Among the three types of relationships, spatial and semantic relationships can be grouped as explicit relationships.

**Relation-Aware Graph Attention Network.** As shown in Fig.5, we construct the multi-relationship graph for both main and reference images, and use the relation-aware graph attention network (ReGAT) proposed by (Li et al., 2019) to learn the graph representation for each image, and embed the image into the final latent feature. In a relation-aware graph attention network, edge labels are embedded to calculate the attention weights between nodes. Please refer to Appendix. A.4 for details of the calculation. For simplicity, we use $G_{spa}(\cdot), G_{sem}(\cdot),$ and $G_{imp}(\cdot)$ to represent the spatial graph module, the semantic graph module, and the implicit graph module, respectively.

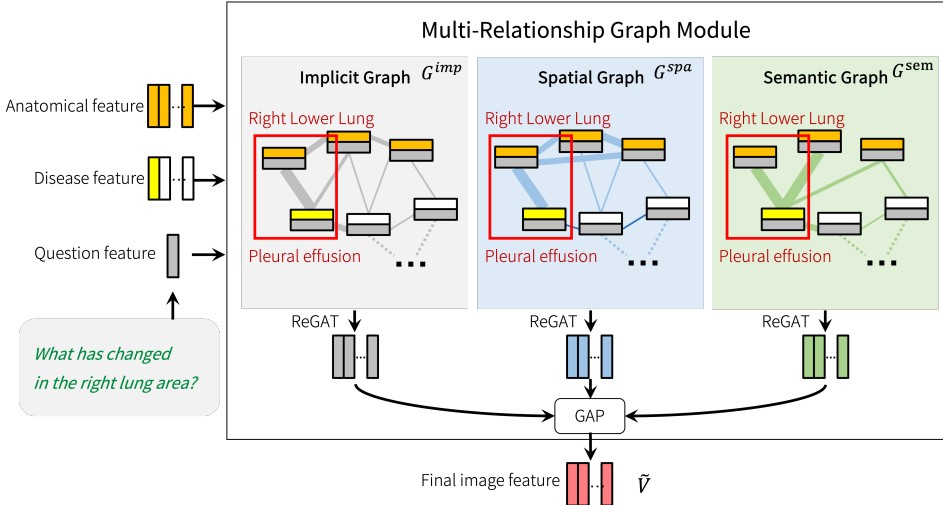

Figure 5: Multi-modal relationship graph module.

Given the input feature nodes $\mathbf{V}$ of each image, the final graph feature $\widetilde{V}$ can be represented as:

$$\widetilde{\mathbf{V}} = GAP(G_{spa}(\mathbf{V}) + G_{sem}(\mathbf{V}) + G_{imp}(\mathbf{V})) \tag{1}$$

where $GAP(\cdot)$ means the global average pooling. The image difference graph features $\widetilde{\mathbf{V}}^{diff}$ is constructed by subtracting the node feature and edge feature between the main and reference image:

$$\widetilde{\mathbf{v}}_i^{diff} = \widetilde{\mathbf{v}}_i^{main} - \widetilde{\mathbf{v}}_i^{ref}, i = 1, \cdots, 2N, \tag{2}$$

where $\widetilde{\mathbf{v}}_i^{diff}, \widetilde{\mathbf{v}}_i^{main}, \widetilde{\mathbf{v}}_i^{ref} \in \mathbb{R}^d$ represent the final feature for the $i$-th node of graphs. Therefore, the final graph features $\widetilde{\mathbf{V}}^{diff}, \widetilde{\mathbf{V}}^{main}, \widetilde{\mathbf{V}}^{ref} \in \mathbb{R}^{2N \times d}$ can be obtained.

**Feature Attention and Answer Generation** Following previous work (Tu et al., 2021), the generated main, reference, and difference features $\widetilde{\mathbf{v}}_i^{main}, \widetilde{\mathbf{v}}_i^{ref}, \widetilde{\mathbf{v}}_i^{diff}$ are then fed into the Feature Attention Module, which first calculates the attention weights of each node, then output the final feature vectors $\mathbf{l}_m$, $\mathbf{l}_r$, and $\mathbf{l}_{diff}$. For details of the calculation, please refer to Appendix A.5. Finally, by feeding the final feature vectors $\mathbf{l}_m$, $\mathbf{l}_r$, and $\mathbf{l}_{diff}$ into the Answer Generation module, the final answer is generated. Same as (Tu et al., 2021)'s setting, the Answer Generation module is composed of LSTM networks and attention modules. The Part-Of-Speech (POS) information is also considered to help generate the answers. For the calculation details, please also refer to Appendix A.6 . We adopt the generative language model because our questions have highly diverse answers. (e.g. the *difference* type question). A simple classification model is not adequate for our task.

## 3 EXPERIMENTS

**Datasets.** *MIMIC-CXR.* The MIMIC-CXR dataset is a large publicly available dataset of chest radiographs with radiology reports, containing 377,110 images corresponding to 227,835 radiograph studies from 65,379 patients (Johnson et al., 2019). One patient may have multiple studies, and each study consists of a radiology report and one or more images. Two primary sections of interest in reports are findings: a natural language description of the important aspects of the image and an impression: a short summary of the most immediately relevant findings. Our MIMIC-Diff-VQA is constructed based on the MIMIC-CXR dataset.

*Chest ImaGenome.* MIMIC-CXR has been added more annotations by (Wu et al., 2021; Goldberger et al., 2000) including the anatomical structure bounding boxes. This new dataset is named Chest ImaGenome Dataset. We trained the Faster-RCNN to detect the anatomical structures on their gold standard dataset, which contains 26 anatomical structures.

*VinDr.* The VinDr dataset consists of 18,000 images manually annotated by 17 experienced radiologists (Nguyen et al., 2020). Its images have 22 local labels of boxes surrounding abnormalities and six global labels of suspected diseases. We used it to train the pre-trained disease detection model.

**Baselines** Since we are the first to propose this medical imaging difference VQA problem, we have to choose two baseline models from the traditional medical VQA task and image difference captioning task, respectively. One is Multiple Meta-model Quantifying (MMQ) proposed by (Do et al., 2021). The other is Multi-Change Captioning transformers (MCCFormers) proposed by (Qiu et al., 2021).

1.*MMQ* is one of the recently proposed methods to perform the traditional medical VQA task with excellent results. MMQ adopted Model Agnostic Meta-Learning (MAML) (Finn et al., 2017) to handle the problem of the small size of the medical dataset. It also relieves the problem of the difference in visual concepts between general and medical images when finetuning.

2.*MCCFormers* is proposed to handle the image difference captioning task. It achieved state-of-the-art performance on the CLEVR-Change dataset (Park et al., 2019), which is a famous image difference captioning dataset. MCCFormers used transformers to capture the region relationships among intra- and inter-image pairs.

3.*IDC (Yao et al., 2022)* is the state-of-the-art method performed on the general image difference captioning task. They used the pretraining technique to build the bridge between vision and language, allowing them to align large visual variance between image pairs and greatly improve the performance on the challenging image difference dataset, Birds-to-Words (Forbes et al., 2019).

**Results and Discussion.** We implemented the experiments on the PyTorch platform. We used an Adam optimizer with a learning rate of 0.0001 to train our model for 30,000 iterations at a batch size of 64. The experiments are conducted on two GeForce RTX 3090 cards with a training time of 3 hours and 49 minutes. The bounding box feature dimension is 1024. Each word is represented by a 600-dimensional feature vector including a 300-dimensional Glove (Pennington et al., 2014) embedding. We used BLEU (Papineni et al., 2002), which is a popular metric for evaluating the generated text, as the metric in our experiments. We obtain the results using Microsoft COCO Caption Evaluation (Chen et al., 2015). For the comparison with MMQ, we use accuracy as the metric.

**Ablation Study.** In Tab. 2 We present quantitative results of ablation studies of our method with different graph settings, including implicit graph-only, spatial graph-only, semantic graph-only, and full model with all three graphs. The studies were performed on our constructed MIMIC-Diff-VQA dataset. Although the overall gain on metrics is slight, we visualized the ROIs of our model using different graphs in Appendix A.8 to demonstrate the interpretability gain in some specific question types, such as the questions related to location, and semantic relationships between abnormalities.

Table 2: Quantitative results of our model with different graph settings performed on the MIMIC-Diff-VQA dataset

| Metrics | Implicit | Spatial | Semantic | Full |
|---------|----------|---------|----------|----------|
| Bleu-1 | 0.627 | 0.616 | 0.616 | **0.630** |
| Bleu-2 | 0.543 | 0.530 | 0.533 | **0.546** |
| Bleu-3 | 0.480 | 0.464 | 0.469 | **0.482** |
| Bleu-4 | 0.424 | 0.408 | 0.415 | **0.426** |

**Comparison of accuracy.** Due to the nature of MMQ being a classification model, MMQ is unable to perform on our *difference* question type because of the diversity of answers. Also, given that the baseline model cannot take in two images simultaneously, we excluded the *difference* type question from this comparison. Therefore, we compare our method with MMQ only on the other six types of questions, including *abnormality, presence, view, location, type*, and *level*. These six types of questions have a limited number of answers. In order to compare with them, we use accuracy as the metric for comparison. Please note that our method is still a text-generation model. We count the predicted answer as a True answer only when the prediction is fully matched with the ground truth answer.

The comparison results are shown in Tab. 3. We have refined the comparison more into open-ended question results and closed-ended question (with only 'yes' or 'no' answers) results. It is clear from the results that the current VQA model has difficulty handling our dataset because of the lack of focus on the key regions and the ability to find the relationships between anatomical structures and diseases. Also, even after filtering out the *difference* questions, there are still 9,231 possible answers in total. It is difficult for a classification model to localize the optimal answer from such a huge amount of candidates.

Table 3: Accuracy comparison between our method and MMQ.

| Question | Open | Closed | Total |
|---|---|---|---|
| MMQ | 11.5 | 10.8 | 11.5 |
| **Ours** | 25.59 | 74.20 | 49.29 |

**Comparison of quality of the text.** For the *difference* question, we use the metrics for evaluating the generated text. The comparison results between our method, MCCFormers, and IDC are shown in Tab. 4. Our method significantly outperforms MCCFormers on every metric. IDC performs better but is still not comparable to ours. The CIDEr (Vedantam et al., 2015) metric, a measure of similarity between sentences, even reached 0 on MCCFormers, which means it failed to provide any meaningful keywords in the answers. This is because the generated answers of MCCFormers are almost identical, and it failed to identify the differences between images. Although MCCFormers is a difference captioning method, it compares patch to patch directly. It may work well in the simple CLVER dataset. However, when it comes to medical images, most of which are not aligned well, the patch-to-patch method cannot identify which region corresponds to a specific anatomical structure. Furthermore, MCCFormers requires no medical knowledge graphs to find the relationships between different regions. IDC has the ability to align significant variances between images. This enables them to have much higher results than MCCFormers. However, they still use pre-trained patch-wise image features, which is not feasible in the medical domain with more fine-grained features.

Table 4: Comparison results between our method and MCCFormers on *difference* questions of the MIMIC-diff-VQA dataset

| Metrics | MCCFormers | IDC | Ours |
|---|---|---|---|
| Bleu-1 | 0.214 | 0.525 | **0.641** |
| Bleu-2 | 0.190 | 0.464 | **0.564** |
| Bleu-3 | 0.170 | 0.405 | **0.500** |
| Bleu-4 | 0.153 | 0.354 | **0.441** |

**Visualization.** Visualized results can be found in Appendix A.8.

## 4 CONCLUSION

First, We proposed a medical image difference VQA problem and constructed a large-scale MIMIC-Diff-VQA dataset for this task, which is valuable to both the research and medical communities. Also, we designed an anatomical structure-aware multi-relation image difference graph to extract image-difference features. We trained an image difference VQA framework utilizing medical knowledge graphs and compared it to current state-of-the-art methods with improved performances. However, our constructed dataset is currently only focusing on the common cases and ignoring special ones—for example, cases where the same disease appears in more than two places. Our current Key-Info dataset can only take care of, at most, two locations of the same disease. Future work could be extending the dataset to consider more special cases.

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

## A  APPENDIX

### A.1  RELATED WORK

**Image Difference Captioning.** The analysis of different images has been explored by a number of researchers in the general domain. The exploration of Image Difference Captioning(IDC) can be split into three stages. The beginning of the first stage is marked by the "spot-the-diff" dataset (Jhamtani & Berg-Kirkpatrick, 2018), which consists of different frames of the same video surveillance footage. This is also the very first time that the IDC task has been proposed. In this phase, the researchers only focus on the pixel-level difference in the same view of the same scene. (Jhamtani & Berg-Kirkpatrick, 2018) use the clusters of differing pixels as a proxy for exposing object-level differences. (Tan et al., 2019; Oluwasanmi et al., 2019) propose to employ encoder-decoder architecture with attention modules to find the relationship between two images. In the second phase, the challenge was upgraded by adding different view angles of the scenes. This demands a higher requirement for the analysis of different regions between images. The iconic dataset in this phase is the CLEVR-change dataset (Park et al., 2019), which comprises pictures of a group of objects(cube, sphere, and cylinder) from different views. The attention mechanism is widely employed to address this challenge (Park et al., 2019; Shi et al., 2020; Tu et al., 2021; Sun et al., 2022; Kim et al., 2021; Qiu et al., 2021). (Hosseinzadeh & Wang, 2021) propose to use an auxiliary task to enhance the primary task to generate the captions. (Liao et al., 2021) consider 3D information and adopt a scene graph to assist in localizing the changing objects. (Kim et al., 2021) also introduces a CLEVR-DC dataset, which is similar to CLEVR-change, but with a larger viewpoint change. In the third phase, more fine-grained visual differences are shown in the image pairs. The Birds-to-Words dataset (Forbes et al., 2019) is composed of a variety of bird images, and each image pair is captioned by human observers. Since the species, posture, and background of the birds in each picture vary greatly, this desires a new method to solve the problem. (Forbes et al., 2019) proposed Neural Naturalist, which is a transformer-based model. (Yan et al., 2021) learns to understand the semantic structures while comparing the images by leveraging image segmentation with a novel semantic pooling and using graph convolutional networks to perform reasoning. (Yao et al., 2022) embrace the pre-training technique to align the visual difference and the text descriptions and achieve state-of-the-art performance. We compared our method with theirs and outperformed them on our medical image difference dataset.

**Medical Visual Question Answering.** Medical visual question answering aims to answer clinical questions given medical images. Medical images span a wide spectrum of modalities, including CT/MRI imaging, histopathology images, angiography, characteristic imaging appearance, ultrasound, and radiographs (Abacha et al., 2019; Lau et al., 2018; He et al., 2020). Clinical questions mainly ask for modality, plane, organ system, and abnormality (Abacha et al., 2019). However, large and well-annotated medical VQA datasets are still in scarcity. Previous MED-VQA methods mostly employ a two-stage procedure: 1) extract visual features on medical images through a detection model like Faster-RCNN (Ren et al., 2015), YOLO (Redmon et al., 2016), and extract question features via BERT (Devlin et al., 2018); 2) attempt to aggregate visual and question features for predicting the final answer (Zhan et al., 2020; Abacha et al., 2018; Zhou et al., 2018; Shi et al., 2019; Yan et al., 2019). (Lau et al., 2018) deploys existing VQA models, i.e., the stacked attention network

(SAN) (Yang et al., 2016) and the multimodal compact bilinear pooling (MCB) (Fukui et al., 2016), in general domains to solve MED-VQA. (Nguyen et al., 2019) proposes to mix enhanced visual features framework with different attention mechanisms such as bilinear attention network (BAN) (Kim et al., 2018) and SAN. (Zhan et al., 2020) proposes separate reasoning modules for different questions to improve the reasoning on medical questions. (Shi et al., 2019) integrates question categories and question topic distributions to assist answer prediction. (Yan et al., 2019) improves the CNN feature extractor with global average pooling to boost classification. (Zhou et al., 2018) applies some image enhancement methods by reconstructing with small random rotations, offsets, scaling, and clipping to boost classification. However, the MED-VQA problem still suffers from lacking fine-grained annotations on images, massive diversity of medical data types, and medical reasoning skills from professions, and is thus far from practical.

**Other related work.** In the general domain, NS-VQA (Yi et al., 2018) proposed to extract regions of interest(ROIs) with predicted semantic labels and generate scene graphs based on the semantic labels using Mask-RCNN. However, NS-VQA focused on leveraging pre-designed python logical programs to process different questions and interpret(calculate) the answers. NS-VQA's answer generation greatly relies on the quality of the object segmentation and labeling by pre-trained Mask-RCNN. Since NS-VQA only evaluated the performance on a simple dataset: CLVER, where all pictures have a single color background, each object has a fixed number of labels and the same label types. Thus, training Mask-RCNN to detect different objects on this dataset is easy to obtain an ideal performance.

(Liu et al., 2021) proposed to extract abnormality-related image features by constructing a pool of normal chest x-ray images and using contrastive learning to distill the contrastive features between abnormal and normal images to improve the report generation performance. However, We focus on comparing the past visiting and current visiting images from the same patient to track the subtle changes that happened between the two visits. Our method is clinically driven and aims at helping the radiologist validate the hypothesis of what has changed after the intervention for each patient.

## A.2 MIMIC-DIFF-VQA DATASET CONSTRUCTION

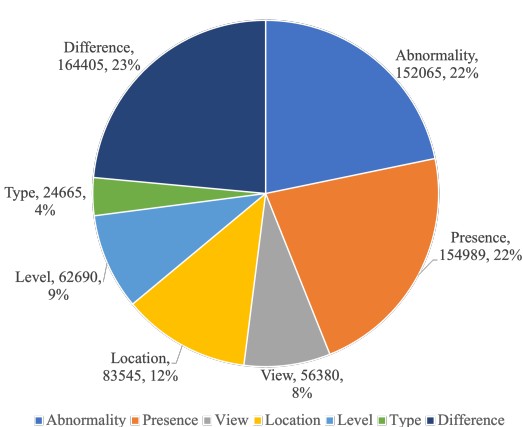

Figure 6: Statistics by question types

First, we exclude the lateral view and select only the common PA or AP views for comparison. Patients with only one radiology visit are also excluded because no second image is available for differential comparison.

Next, we collect a set of abnormality names, as well as the sets of important attributes including location, level, and type, from the filtered MIMIC-CXR dataset. The lists of abnormality names and the attribute words are collected by iteratively extracting entities from random reports using ScispaCy (Neumann et al., 2019), which is a SpaCy model for biomedical text processing. Then we manually go through all the extracted entities that haven't been added to the collection list and select the common keywords that appear frequently. Then we add these selected keywords to the collection

Table 5: Full list of examples for each question type.

| Question type | example |
|---|---|
| Abnormality | what abnormalities are seen in this image?
what abnormalities are seen in the *[location]*?
is there evidence of any abnormalities in this image?
is this image normal? |
| Presence | is there evidence of *[abnormality]* in this image?
is there *[abnormality]*?
is there *[abnormality]* in the *[location]*? |
| View | which view is this image taken?
is this PA view?
is this AP view? |
| Location | where in the image is the *[abnormality]* located?
where is the *[abnormality]*?
is the *[abnormality]* located on the left side or right side?
is the *[abnormality]* in the *[location]*? |
| Level | what level is the *[abnormality]*? |
| Type | what type is the *[abnormality]*? |
| Difference | what has changed compared to the reference image?
what has changed in the *[location]* area? |

lists of abnormality names and attributes. During this process, different variants that represent the same abnormality are also recorded. Next, for each study, we use regular expressions to localize the abnormality names as well as their variants to detect attribute words near these detected abnormalities. (Here, "study" represents a single patient visit. Please refer to Section 3 for more context.) Meanwhile, by going through the extracted entities, we manually select the keywords/expressions that indicate negation information to localize the negative findings, i.e. cases where the abnormality does not exist. After updating the keyword lists, we keep repeating this Extract-Check-Fix cycle until minimum mistakes are found.

Thereafter, a dataset of single studies can be constructed accordingly. We call this dataset the Key-Info dataset. As shown in Fig. 7, for each study, the Key-Info dataset provides information on every positive finding and its corresponding attributes as well as the negative findings. The full lists of the selected abnormality names and the attribute words are shown in Tab. 6 and Tab. 7, respectively. The "posterior location" attribute represents the location information that appears after the abnormality keyword in a sentence.

**Study pairing and question generation**

When the abnormality database is constructed, questions for study pairs can be generated accordingly. The examples of each question type are shown in Tab. 1. Each image pair contains the main image and a reference image, which are extracted from different studies. Among all the question types, the first six question types are for the main image only, and the *difference* question is for both images.

A.2.1    DATASET VALIDATION

To further verify the reliability of our constructed dataset, 3 human verifiers were assigned 1200 random sampled question-answer pairs along with the reports and evaluated each sample by annotating "correct" or "incorrect". Finally, the accuracy of the evaluation achieved 97.33%, which is acceptable for training neural networks. Tab. 8 shows the evaluation results of each verifier. It proves that our approach of constructing a dataset in an Extract-Check-Fix cycle works well in ensuring that the constructed dataset has minimum mistakes.

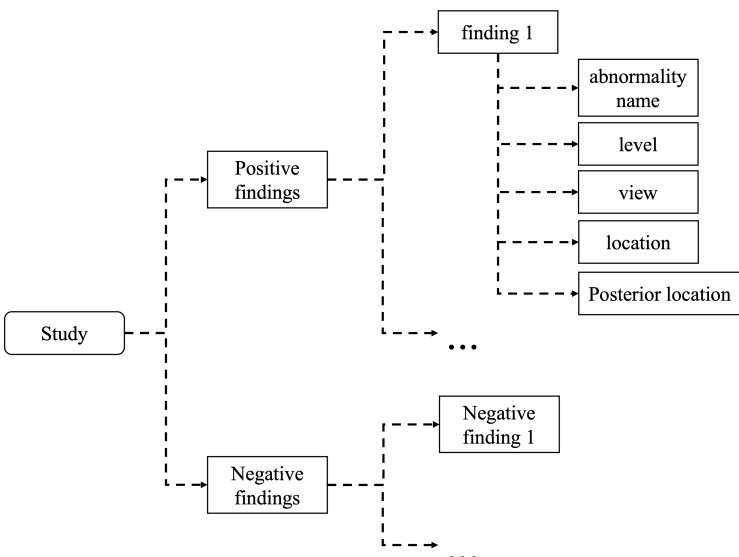

Figure 7: Structure of one study in the Key-Info dataset

## A.3 ANATOMICAL STRUCTURE DETECTION

The Anatomical structure detection results are shown in Tab. 9. We also tested our trained Faster-RCNN on examples of diffuse and non-diffuse diseases to verify the robustness of our detection model. We select interstitial edema as the diffuse disease. Diffuse diseases accounted for 5 out of a total of 200 examples, and non-diffuse diseases accounted for 195 examples.

## A.4 RELATION-AWARE GRAPH ATTENTION NETWORK

For the implicit relationship, each updated node $\widetilde{\mathbf{v}}_i \in \mathbb{R}^d$ in the final graph can be calculated as below:

$$\widetilde{\mathbf{v}}_i = \mathbf{W}^o \cdot (\|_{m=1}^M \sigma(\sum_{j \in \mathcal{N}_i} \alpha_{ij} \mathbf{W}^m \mathbf{v}_j)) \tag{3}$$

where $\mathcal{N}_i$ is the neighborhood set of the node $i$, $\mathbf{W}^m \in \mathbb{R}^{d \times (d_f + d_q)}$ is the projection matrix, $d$ is the dimension of the final node feature, $\sigma$ is the activation function, $\|_{m=1}^M$ represents concatenating the output of the $M$ attention heads, $\mathbf{W}^o \in \mathbb{R}^{d \times Md}$. The attention weights $\alpha_{ij}$ between the node $i$ and node $j$ consider the similarity between node pairs and the relations between the corresponding region locations. The calculation for $\alpha_{ij}$ can be formulated as:

$$\alpha_{ij} = \frac{\alpha_{ij}^b \cdot \exp\left(\alpha_{ij}^v\right)}{\sum_{j=1}^K \alpha_{ij}^b \cdot \exp\left(\alpha_{ij}^v\right)} \tag{4}$$

$$\alpha_{ij}^v = (\mathbf{U}\mathbf{v}_i)^\top \cdot (\mathbf{V}\mathbf{v}_j) \tag{5}$$

$$\alpha_{ij}^b = \max\left(0, w \cdot f_b(\mathbf{b}_{ij})\right) \tag{6}$$

where $U, V \in \mathbb{R}^{d \times (d_f + d_q)}$ are projection matrices. $\mathbf{b}_{ij}$ is the relative geometry feature between node $i$ and $j$, and can be calculated by $[\log(\frac{|x_i - x_j|}{w_i}), \log(\frac{|y_i - y_j|}{h_i}), \log(\frac{w_j}{w_i}), \log(\frac{h_j}{h_i})]$, $f_b$ is a function that embeds the 4-dimensional relative geometry feature into $d$-dimensional, $w \in \mathbb{R}^d$ is a vector that transforms the feature into a scalar weight. The bounding box coordinates, widths, and heights of the node $i$ and $j$ can be represented by $x_i, x_j, y_i, y_j, w_i, w_j, h_i$, and $h_j$.

Spatial and semantic graphs, which can also be called explicit graphs, can be seen as directed graphs. The updating rule considers the relation directions between node pairs and the labels of the edges. The formulation of a single attention head is shown below:

Table 6: Applicable disease names

| id | Disease names |
| --- | --- |
| 0 | pleural effusion |
| 1 | atelectasis |
| 2 | cardiomegaly |
| 3 | enlargement of the cardiac silhouette |
| 4 | edema |
| 5 | hernia |
| 6 | vascular congestion |
| 7 | hilar congestion |
| 8 | pneumothorax |
| 9 | heart failure |
| 10 | lung opacity |
| 11 | pneumonia |
| 12 | tortuosity of the descending aorta |
| 13 | scoliosis |
| 14 | gastric distention |
| 15 | hypoxemia |
| 16 | hypertensive heart disease |
| 17 | hematoma |
| 18 | tortuosity of the thoracic aorta |
| 19 | contusion |
| 20 | emphysema |
| 21 | granuloma |
| 22 | calcification |
| 23 | pleural thickening |
| 24 | thymoma |
| 25 | blunting of the costophrenic angle |
| 26 | consolidation |
| 27 | fracture |
| 28 | pneumomediastinum |
| 29 | air collection |

$$\widetilde{\mathbf{v}}_i = \sigma\Big(\sum_{j\in\mathcal{N}_i} \alpha_{ij}\mathbf{W}_{dir(i,j)}\mathbf{v}_j + b_{lab(i,j)}\Big) \tag{7}$$

$$\alpha_{ij} = \frac{\exp\left((\mathbf{U}\mathbf{v}_i)^\top \cdot \mathbf{V}_{dir(i,j)}\mathbf{v}_j + c_{lab(i,j)}\right)}{\sum_{j\in\mathcal{N}_i}\exp\left((\mathbf{U}\mathbf{v}_i)^\top \cdot \mathbf{V}_{dir(i,j)}\mathbf{v}_j + c_{lab(i,j)}\right)} \tag{8}$$

where $dir(i,j)$ represents the direction goes from node $i$ to $j$, $lab(i,j)$ is the label assigned to the edge $(i,j)$, $W_{dir(i,j)}, V_{dir(i,j)} \in \mathbb{R}^{d\times(d_f+d_q)}$ are projection matrices, $b_{lab(i,j)}, c_{lab(i,j)} \in \mathbb{R}^d$ are bias terms. The multi-head attention can be calculated similarly by concatenating the output features and adding a projection matrix $\mathbf{W}^o \in \mathbb{R}^{d\times Md}$.

### A.5 FEATURE ATTENTION MODULE

The generated main image features $\widetilde{\mathbf{V}}_i^{main}$, reference image feature $\widetilde{\mathbf{V}}_i^{ref}$ and the difference feature $\widetilde{\mathbf{V}}_i^{diff}$ are then fed into the Feature Attention Module, which is similar to the two modules in (Tu et al., 2021) called Cross-semantic Relation Measuring block(CSRM) and Prior Knowledge-guided Change Localizer. In the Feature Attention module, we first calculate the prior knowledge $C'_m$, and $C'_r$ for the main image and the reference image, respectively. Take $C'_m$ for example, the calculation process is shown below.

$$C_m = \phi(\widetilde{\mathbf{V}}^{main}W_q^c + \widetilde{\mathbf{V}}^{main}W_v^c + b^c) \tag{9}$$

$$A_m = \sigma(\widetilde{\mathbf{V}}^{main}W_q^a + \widetilde{\mathbf{V}}^{main}W_v^a + b^a) \tag{10}$$

Table 7: Attribute keywords for level, location(pre), location(post), and type.

| Attribute | | | |
|---|---|---|---|
| level | location(pre) | location(post) | type |
| moderate | mid to lower | the lower lobe | interstitial |
| acute | left | the upper lobe | layering |
| mild | right | the middle lobe | dense |
| small | retrocardiac | the left lung base | parenchymal |
| moderately | pericardial | the right lung base | compressive |
| severe | bibasilar | the lung bases | obstructive |
| moderate to large | bilateral | the left base | linear |
| moderate to severe | basilar | the right base | plate-like |
| mild to moderate | apicolateral | the right upper lung | patchy |
| moderate to large | basal | the left upper lung | ground-glass |
| minimal | left-sided | the right middle lung | calcified |
| mildly | lobe | the left middle lung | scattered |
| subtle | lung | the right mid lung | interstitial |
| massive | area | the left mid lung | focal |
| minimally | right-sided | the right lower lung | multifocal |
| increasing | apical | the left lower lung | multi-focal |
| decreasing | pleural | the right upper lobe | loculated |
| minor | upper | the left upper lobe | hazy |
| trace | lower | the right middle lobe | |
| | middle | the left middle lobe | |
| | mid | the right mid lobe | |
| | rib | the left mid lobe | |
| | | the right lower lobe | |
| | | the left lower lobe | |
| | | the left apical area | |
| | | the left apical region | |
| | | the right apical area | |
| | | the right apical region | |
| | | the apical region | |
| | | the apical area | |
| | | the right mid to lower lung | |
| | | the left mid to lower lung | |
| | | the medial right lung base | |
| | | the medial left lung base | |
| | | the upper lungs | |
| | | the lower lungs | |
| | | the upper lobes | |
| | | the lower lobes | |
| | | the right mid to lower hemithorax | |
| | | the soft tissues | |
| | | the right midlung | |
| | | the left midlung | |

$$C'_m = A_m \odot C_m \qquad (11)$$

where $C_m \in \mathbb{R}^{2N \times d}$ is the "candidate change", $A_m \in \mathbb{R}^{2N \times d}$ is the "attention gate", $W_q^c, W_v^c, W_q^a, W_v^a \in \mathbb{R}^{d \times d}$, $b^c, b^a \in \mathbb{R}^d$, $\odot$ represents the element-wise multiplication, $\phi$ is the tanh function, $\sigma$ is the sigmoid function. $C'_r$ can be calculated similarly.

Table 8: Evaluation results by human verifiers(todo)

| Verifier | # of examples | # of correctness | Accuracy |
|---|---|---|---|
| Verifier 1 | 500 | 475 | 95% |
| Verifier 2 | 1000 | 989 | 98.9% |
| Verifier 3 | 200 | 193 | 96.5% |
| Total | 1700 | 1657 | 97.4% |

Table 9: Anatomical structure detection results. Precision represents when the Intersection over Union(IoU) threshold is set to 0.5.

| Category | Precision (IoU =0.5) | Diffuse disease Precision | non-diffuse Precision |
|---|---|---|---|
| right lung | 97.561 | 100 | 97.569 |
| right lower lung zone | 88.774 | 100 | 88.72 |
| right costophrenic angle | 68.294 | 80.198 | 68.178 |
| left upper lung zone | 95.075 | 100 | 95.114 |
| left hilar structures | 90.092 | 100 | 90.479 |
| left hemidiaphragm | 76.314 | 72.277 | 76.908 |
| left clavicle | 83.859 | 100 | 83.808 |
| svc | 87.734 | 100 | 87.729 |
| right atrium | 80.54 | 100 | 80.457 |
| right upper lung zone | 95.55 | 100 | 95.562 |
| right hilar structures | 92.887 | 100 | 92.877 |
| right hemidiaphragm | 83.766 | 100 | 83.7 |
| left mid lung zone | 87.251 | 100 | 87.774 |
| left apical zone | 92.654 | 100 | 93.312 |
| trachea | 89.421 | 100 | 89.444 |
| aortic arch | 90.951 | 100 | 90.957 |
| cardiac silhouette | 90.643 | 100 | 90.812 |
| carina | 45.423 | 30.693 | 45.821 |
| right mid lung zone | 91.776 | 100 | 91.754 |
| right apical zone | 93.352 | 100 | 93.354 |
| left lung | 96.695 | 100 | 96.942 |
| left lower lung zone | 82.534 | 100 | 83.01 |
| left costophrenic angle | 63.95 | 80.198 | 64.321 |
| right clavicle | 87.384 | 100 | 87.393 |
| upper mediastinum | 95.216 | 100 | 95.26 |
| cavoatrial junction | 66.503 | 100 | 65.747 |

Then, guided by the prior knowledge, we calculate the attention weights $a_m$ and $a_r$ for the main image and the reference image respectively. The formulations are shown below:

$$a_m = \sigma(\text{FC}_2(\text{ReLU}(\text{FC}_1([\widetilde{\mathbf{V}}^{main}; \widetilde{\mathbf{V}}^{diff}; C'_m])))) \tag{12}$$

$$a_r = \sigma(\text{FC}_2(\text{ReLU}(\text{FC}_1([\widetilde{\mathbf{V}}^{ref}; \widetilde{\mathbf{V}}^{diff}; C'_r])))) \tag{13}$$

where $[;]$ represents the concatenation, $FC$ represents fully-connected layer, $\sigma$ represents the sigmoid function.

After obtaining the attention weights $a_m \in \mathbb{R}^{2N}$ and $a_r \in \mathbb{R}^{2N}$, the final image feature vector $\mathbf{l}_m$ and $\mathbf{l}_r$ for the main image and the reference image can be calculated as follows:

$$\mathbf{l}_m = \sum_{i=1}^{2N} a_{m_i} \widetilde{\mathbf{v}}_i^{main} \tag{14}$$

$$\mathbf{l}_r = \sum_{i=1}^{2N} a_{r_i} \widetilde{\mathbf{v}}_i^{ref} \tag{15}$$

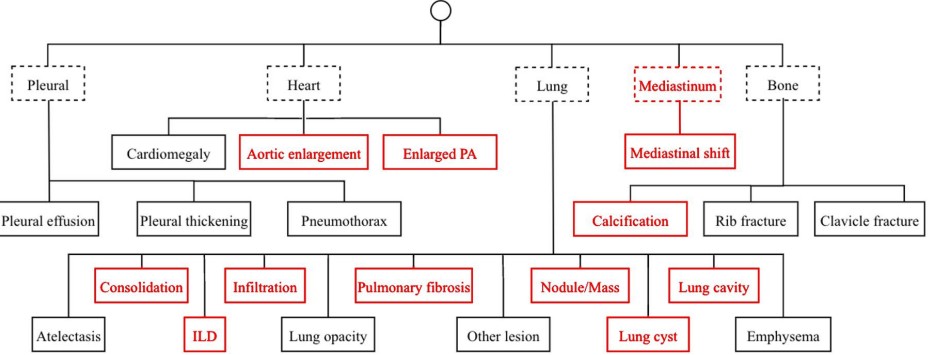

(a) Anatomical knowledge graph (compared to Zhang et al. (2020), our new added disease types are annotated by red.)

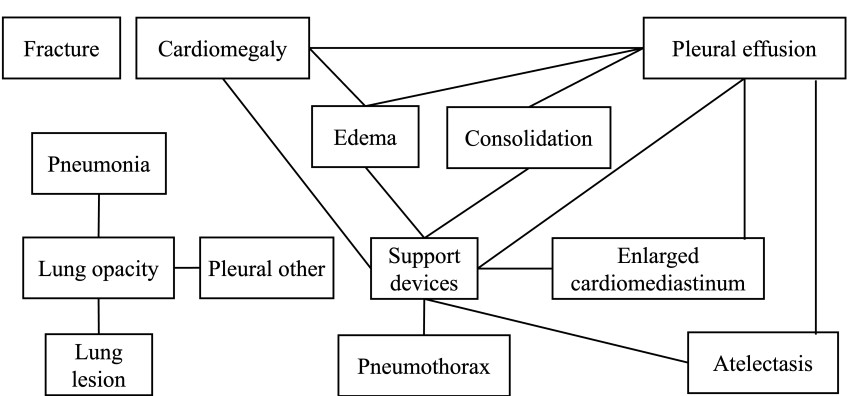

(b) Co-occurance knowledge graph

Figure 8: Knowledge graphs

where $a_m \in \mathbb{R}^{2N}$ and $a_r \in \mathbb{R}^{2N}$ are the attention weights. The difference vector is accordingly computed as:

$$\mathbf{l}_{diff} = \mathbf{l}_m - \mathbf{l}_r \tag{16}$$

### A.6 ANSWER GENERATION

**Dynamic Feature Generation.** At each time step $t$, we first calculate the attention weights $\alpha_i^{(t)}$, which is for calculating the intermediate dynamic feature $l_{dyn}^{(t)}$ in the next step. The $\alpha_i(t)$ can be calculated as follows:

$$v = \text{ReLU}(W_{a_1}[l_{bef}; l_{diff}; l_{aft}] + b_{a_1}) \tag{17}$$
$$u^{(t)} = [v; h_c^{(t-1)}] \tag{18}$$
$$h_a^{(t)} = LSTM_a(h_a^{(t)}|u^{(t)}, h_a^{(0:t-1)}) \tag{19}$$
$$\alpha_i^{(t)} \sim Softmax(W_{a_2}h_a^{(t)} + b_{a_2}) \tag{20}$$

where $W_{a_1}, W_{a_2}, b_{a_1}, b_{a_2}$ are learnable parameters, $LSTM_a$ is a LSTM network used as attention weights generator, $h_a^{(t)}$ is the output of the $LSTM_a$ at the time step $t$, $h_c^{(t-1)}$ is the output of the answer generator $LSTM_c$ at the time step $t-1$, which will be explained in more detail later.

Then, the intermediate dynamic feature $l_{dyn}^{(t)}$ can then be calculated as follows:

$$l_{dyn}^{(t)} = \sum_i \alpha_i^{(t)} l_i \tag{21}$$

where $i \in (bef, diff, aft)$.

Before calculating the final dynamic feature $L_{dyn}^{(t)}$, POS feature $p^{(t)}$ needs to be obtained first. The POS feature is calculated from the hidden embedding of the answer $h_c^{(t-1)}$ from the last time step. The calculation can be formulated as below:

$$h_p^{(t)} = \text{ReLU}(W_{p_1} h_c^{(t-1)} + b_{p_1}) \tag{22}$$

$$w_p^{(t)} = Softmax(W_{p_2} h_p^{(t)} + b_{p_2}) \tag{23}$$

$$p^{(t)} = E_p w_p^{(t)} \tag{24}$$

where $W_{p_1}, W_{p_2}, b_{p_1}, b_{p_2}$ are learnable parameters, $E_p$ is a learnable POS embedding matrix.

With the intermediate dynamic feature $l_{dyn}^{(t)}$ and the POS feature $p^{(t)}$, we can calculated the final dynamic feature $L_{dyn}^{(t)}$.

$$\beta_t = \sigma(W_{c_2}(\text{ReLU}(W_{c_1}[p^{(t)}; h_c^{(t-1)}; l_{dyn}^{(t)}]))) \tag{25}$$

$$L_{dyn}^{(t)} = \beta_t \odot l_{dyn}^{(t)} \tag{26}$$

where the range of $\beta_t$ is $[0, 1]$, the value of it indicates how much the visual information will be used in the answer generation part.

**Answer generator.** The answer is generated by an LSTM network word by word. The initial word at time step 0 is the $< start >$ token.

$$c^{(t)} = [E[w^{(t-1)}]; L_{dyn}^{(t)}] \tag{27}$$

$$h_c^{(t)} = \text{LSTM}_c(h_c^{(t)}|c^{(t)}, h_c^{(0:t-1)}) \tag{28}$$

$$w^{(t)} \sim Softmax(W_c h_c^{(t)} + b_c) \tag{29}$$

where $E$ is a word embedding layer, $E[w^{(t-1)}]$ is the word embedding for the word $w^{(t-1)}$, $W_c, b_c$ are learnable parameters.

We adopt the generative language model because our questions have highly diverse answers. (e.g. the *difference* type question). A simple classification model is not adequate for our task.

## A.7    OTHER RESULTS

We evaluated our proposed multi-relationship graph for the general chest X-ray image classification-based VQA problem (14 diseases) and compared it to state of art method SYSU-HCP (Gong et al., 2021), the best team in the ImageCLEF VQA-Med 2021 task. As shown in Tab. 10, We use AUC as the metric because answering abnormality questions can be considered a multi-label classification problem. Our model achieved significant improvement compared to the state-of-the-art disease classification performance.

We show the results of our model on each question type in Tab. 11. It is worth noting that, Bleu 3 and Bleu 4 tend to have low scores. This is because the answers to most of the questions are short, except for the "difference" questions. For abnormality questions, 72% of the answers have less than or equal to 2 words; for location questions, 79% of the answers have less than or equal to 2 words; 93% of level questions have one-word answers.

## A.8    VISUALIZATIONS

To prove the improvement of the interpretability of our model by adding the spatial and semantic graphs, we visualize the ROIs of our model using different graphs and demonstrate the predictions. As shown in Fig. 9(b), our model using the only implicit graph missed the regions important for the question and failed to interpret the correct answer. In contrast, as shown in Fig. 9(a), with the help

Table 10: Results of classification-based VQA problem.

| Answer | SYSU-HCP | Ours |
|---|---|---|
| Pneumothorax | 0.806 | 0.876 |
| edema | 0.737 | 0.893 |
| lung lesion | 0.665 | 0.843 |
| no | 0.537 | 0.951 |
| lung opacity | 0.605 | 0.859 |
| ateletasis | 0.645 | 0.868 |
| pleural other | 0.858 | 0.845 |
| support devices | 0.769 | 0.924 |
| pneumonia | 0.715 | 0.833 |
| pleural effusion | 0.796 | 0.938 |
| enlarged cardiomediastinum | 0.725 | 0.828 |
| yes | 0.545 | 0.944 |
| consolidation | 0.708 | 0.819 |
| cardiomegaly | 0.688 | 0.892 |
| fracture | 0.664 | 0.871 |
| total (micro) | 0.792 | 0.934 |
| total (macro) | 0.697 | 0.879 |

Table 11: Results of each question type. "-" represents not applicable because no ground truth answer has enough words to trigger the corresponding Bleu metric.

| Question type | Bleu 1 | Bleu 2 | Bleu 3 | Bleu 4 |
|---|---|---|---|---|
| Abnormality | 0.482 | 0.333 | 0.197 | 0.109 |
| Presence | 0.801 | - | - | - |
| View | 0.948 | 0.941 | - | - |
| Location | 0.525 | 0.364 | 0.210 | 0.144 |
| Level | 0.496 | 0.101 | 0.068 | - |
| Difference | 0.641 | 0.564 | 0.500 | 0.441 |

of the spatial relationship graph, our model succeeded in finding the critical region and delivering the correct answer.

Fig. 10 demonstrates a similar scenario on an abnormality-type question. our model using only the implicit graph detected only one abnormality, atelectasis, missed pleural effusion, and lung opacity. However, with the help of the semantic relationship graph, which emphasizes the relationship between pleural effusion, atelectasis, and lung opacity, our full model detected all three abnormalities and provided the correct answer.

**Q:** Is the consolidation located on the left side or right side?
**GT answer:** left side

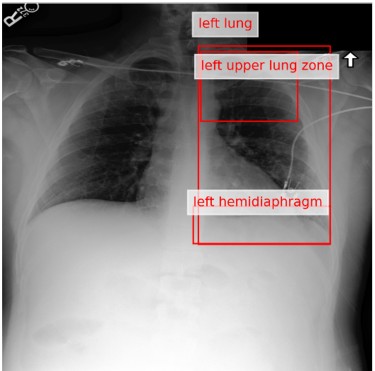
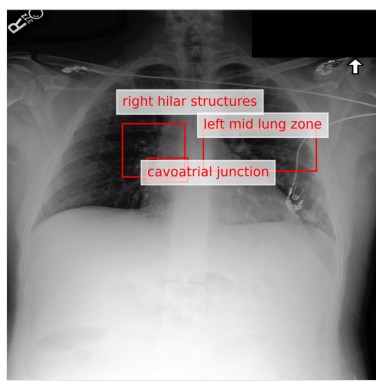

(a) Full model
**Prediction:** left side

(b) Model with implicit graph only
**Prediction:** right side

Figure 9: ROIs Visualization comparison between implicit graph and all graphs on location type question.

**Q:** what abnormalities are seen in this image?
**GT answer:** pleural effusion, atelectasis, lung opacity

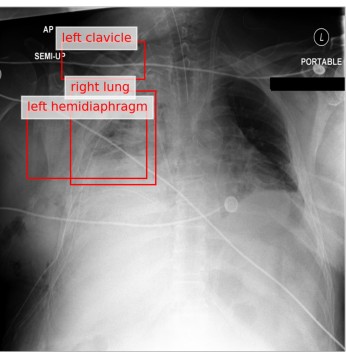
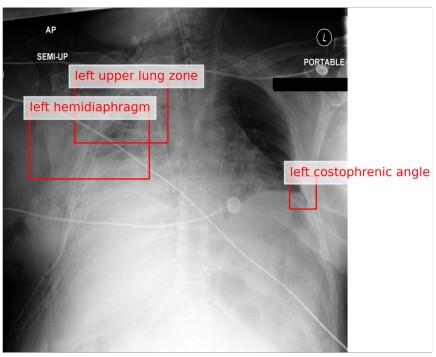

(a) Full model
**Prediction:** pleural effusion,
atelectasis,
lung opacity

(b) Model with implicit graph only
**Prediction:** atelectasis

Figure 10: ROIs Visualization comparison between implicit graph and all graphs on abnormality type question.

As shown in Fig. 11, when asking about pleural effusion, which is an abnormality that happens in the lower lung when there is excess fluid between the layers of the pleura outside the lungs, our method highlighted the corresponding regions (left lower lung). Also, by focusing on these regions, our method can accurately determine the change in the level of pleural effusion between the

main and reference image. In Fig. 12, our method also highlighted cardiac silhouette, this could be because of the strong semantic relationship between cardiomegaly and pleural effusion as mentioned in Section. 2 and Fig. 4a.

Reference                                    Main

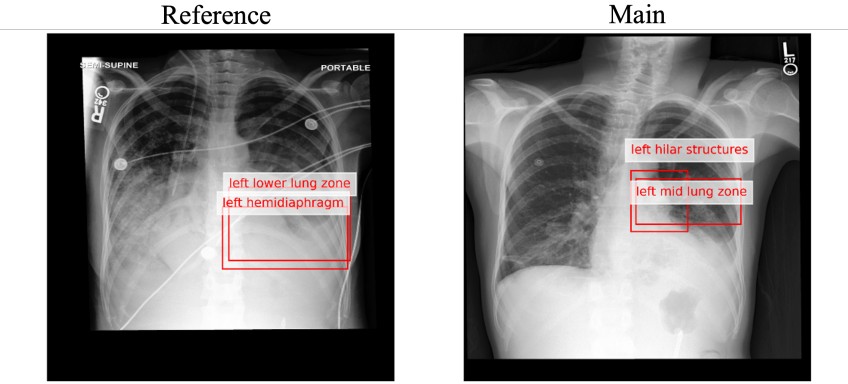

Question: what has changed in the left lung area?

GT Answer: the level of pleural effusion has changed from moderate small to small.

Prediction: the level of pleural effusion has changed from moderate small to small.

Figure 11: Visualization example 1

Reference                                    Main

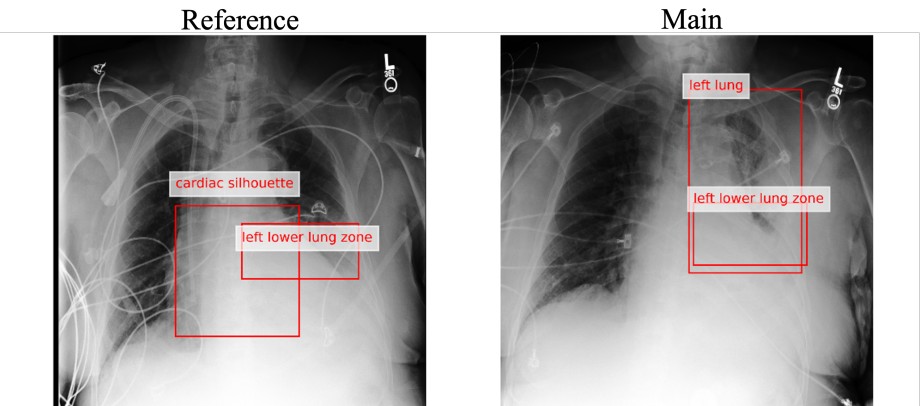

Question: what has changed in the left lung area?

GT Answer: the level of pleural effusion has changed from moderate to small.

Prediction: the level of pleural effusion has changed from moderate to small.

Figure 12: Visualization example 2

