# OpenReview forum: "Anatomical Structure-Aware Image Difference Graph Learning for Difference-Aware Medical Visual Question Answering"
_ICLR.cc/2023/Conference — Submitted to ICLR 2023_

### Official Review · Reviewer_NUDS · 2022-10-13

**Confidence:** 4
**Correctness:** 4
**Technical Novelty And Significance:** 4
**Empirical Novelty And Significance:** 4
**Recommendation:** 8

**Clarity, Quality, Novelty And Reproducibility:**


The introduced dataset is the first of its kind, and the ideas introduced in the model are novel to the best of my knowledge.

The paper is quite clear to read, although I could spot several typos. The citation style used in the paper is often confusing when citations are in the middle of a sentence, as the author's name seems being part of the sentence. For example in page 5 you say: "Each word is tokenized and embedded with Glove Pennington et al. (2014) embeddings.". It would be better to use \citep in latex to get "Each word is tokenized and embedded with Glove (Pennington et al, 2014) embeddings."

**Strength And Weaknesses:**



**STRENGTHS**

1. The authors prepared a large scale dataset that can be used to advance multi-modal models working on medical text/images

2. Most ML applications for chest xray analysis only consider a single chest xray image, which is limiting as lots of medically relevant information can be obtained comparing studies taken at different point in times. This model on the other hand is able to mimic what radiologists do when comparing studies to assess the progression of a disease.

3. How to embed medical domain knowledge in ML models is still a very open research questions. The knowledge-aware graph representation used in this paper is a novel interesting idea.



**WEAKNESSES**
1. Given the heterogeneity of radiological reports, they are quite hard to process with rule-based methods such as the ones used by the authors. I would have like to see a larger validation set than the 100 examples used in the paper.
2. The model relies on anatomical parts segmentation, which is quite challenging in chest x-rays (with large opacities/effusions anatomical parts might not even be visible). What is the classification performance of your anatomical parts classifier? How does it perform on diffuse diseases?
3. Have you tried to use classifiers instead of a language model for some of the tasks 1-6? It would be interesting for example to see a classification baseline for the "abnormality" tasks, that would give an idea of how good your model is.
4. What are the training times of the model?


**Summary Of The Paper:**

This paper introduces a new dataset and visual question answering task to find differences between chest xray images of the same patient taken at two different points in time.

Furthermore, the authors develop a graph representation learning model that uses domain knowledge to solve this task, and provide a baseline for future extensions. This model extracts anatomical/disease features from both images, and passes them as input to a multi-relationship graph module that extracts a final image representation. The features for both images can then be subtracted and used in a language model that generates the final answer.



**Summary Of The Review:**

I believe this paper could be relevant for the ICLR community:
1. The paper introduces a interesting large-scale dataset for vision/language medical tasks.
2. The model developed by the authors is novel.

I therefore argue for acceptance.

---

> ### Author Response · Authors · 2022-11-19
> **Response to Reviewer NUDS (2/2)**
>
> ### Q3. Classification baseline for the "abnormality" tasks.
>
> A3. We have evaluated our proposed multi-relationship graph for the general chest X-ray image classification based on VQA problem (14 diseases) and compared it to state of art methods SYSU-HCP[1]. We added it in Appendix A.7.  Our model achieved significant improvement compared to the state-of-the-art disease classification performance.
>
> We compared our method with SYSU-HCP[1], the best team in the ImageCLEF VQA-Med 2021 task, on the abnormality question type in our constructed dataset. We use AUC as the metric because answering abnormality questions can be considered a multi-label classification problem.
>
>
> > [1] Gong, Haifan, et al. "Sysu-hcp at vqa-med 2021: A data-centric model with efficient training methodology for medical visual question answering." Proceedings http://ceur-ws. org ISSN 1613 (2021): 0073.
>
> | Answer                     | SYSU-HCP | Ours  |
> |----------------------------|----------|-------|
> | Pneumothorax               | 0.806    | 0.876 |
> | edema                      | 0.737    | 0.893 |
> | lung lesion                | 0.665    | 0.843 |
> | no                         | 0.537    | 0.951 |
> | lung opacity               | 0.605    | 0.859 |
> | atelectasis                | 0.645    | 0.868 |
> | pleural other              | 0.858    | 0.845 |
> | support devices            | 0.769    | 0.924 |
> | pneumonia                  | 0.715    | 0.833 |
> | pleural effusion           | 0.796    | 0.938 |
> | enlarged cardiomediastinum | 0.725    | 0.828 |
> | yes                        | 0.545    | 0.944 |
> | consolidation              | 0.708    | 0.819 |
> | cardiomegaly               | 0.688    | 0.892 |
> | fracture                   | 0.664    | 0.871 |
> | total (micro)              | 0.792    | 0.934 |
> | total (macro)              | 0.697    | 0.879 |
>
> ### Q4. What are the training times of the model?
>
> A: We trained on two 3090 cards with 3 hours and 49 minutes for 30,000 iterations.
>
> ### Q5. typos and citations.
> A: We have fixed the citation and typo issues in the revised paper.

---

> ### Author Response · Authors · 2022-11-19
> **Response to Reviewer NUDS (1/2)**
>
> Thank you for your comments and suggestions! We have modified our revised paper accordingly.
>
> ### Q1: more validation of rule based methods.
>
> A1: We have further validated another 1700 examples, and the accuracy reached  97.4% by our rule-based automatic label extraction method. The new result has been updated in the revised paper.
>
> ### Q2. performance of your anatomical parts detector? How does it perform on diffuse diseases?
>
> A:  Our method relies on the detection of anatomical structures, but we do not need to segment the anatomical structures.  The anatomical part detection method performs well on our dataset since we trained this model using the gold standard set of Chest ImaGenome Dataset, with 1000 images and 25967 instances.  In addition, the anatomical structure is relatively larger than disease-related lesions. Therefore, anatomical structure detection is less challenging than disease detection.
>
> We showed the anatomical structure detection Average Precision (AP) with Intersection over union (IOU) equals 0.5. We select interstitial edema as the diffuse disease, the relatively most common diffuse disease in the gold standard dataset in Chest ImaGenome Dataset.
>
> |                          | all 200 examples     | 5 examples                          | 195 examples                    |
> |--------------------------|----------------------|-------------------------------------|---------------------------------|
> | category                 | Precision (iou =0.5) | Diffuse disease Precision (iou=0.5) | non-diffuse Precision (iou=0.5) |
> | right  lung              | 97.561               | 100                                 | 97.569                          |
> | right  lower lung zone   | 88.774               | 100                                 | 88.72                           |
> | right costophrenic angle | 68.294               | 80.198                              | 68.178                          |
> | left upper lung zone     | 95.075               | 100                                 | 95.114                          |
> | left hilar structures    | 90.092               | 100                                 | 90.479                          |
> | left hemidiaphragm       | 76.314               | 72.277                              | 76.908                          |
> | left clavicle            | 83.859               | 100                                 | 83.808                          |
> | svc                      | 87.734               | 100                                 | 87.729                          |
> | right atrium             | 80.54                | 100                                 | 80.457                          |
> | right upper lung zone    | 95.55                | 100                                 | 95.562                          |
> | right hilar structures   | 92.887               | 100                                 | 92.877                          |
> | right hemidiaphragm      | 83.766               | 100                                 | 83.7                            |
> | left mid lung zone       | 87.251               | 100                                 | 87.774                          |
> | left apical zone         | 92.654               | 100                                 | 93.312                          |
> | trachea                  | 89.421               | 100                                 | 89.444                          |
> | aortic arch              | 90.951               | 100                                 | 90.957                          |
> | cardiac silhouette       | 90.643               | 100                                 | 90.812                          |
> | carina                   | 45.423               | 30.693                              | 45.821                          |
> | right mid lung zone      | 91.776               | 100                                 | 91.754                          |
> | right apical zone        | 93.352               | 100                                 | 93.354                          |
> | left lung                | 96.695               | 100                                 | 96.942                          |
> | left lower lung zone     | 82.534               | 100                                 | 83.01                           |
> | left costophrenic angle  | 63.95                | 80.198                              | 64.321                          |
> | right clavicle           | 87.384               | 100                                 | 87.393                          |
> | upper mediastinum        | 95.216               | 100                                 | 95.26                           |
> | cavoatrial junction      | 66.503               | 100                                 | 65.747                          |

---

### Official Review · Reviewer_FdAD · 2022-10-26

**Confidence:** 5
**Correctness:** 2
**Technical Novelty And Significance:** 2
**Empirical Novelty And Significance:** 2
**Recommendation:** 5

**Clarity, Quality, Novelty And Reproducibility:**

The clarity should be improved - there are several typos in the paper. I recommend the author check and revise the text carefully. The novelty is limited: 1) The usefulness of the proposed task should be clarified; 2) An existing work Contrastive Attention in the medical image field and NS-VQA task should be discussed. The reproducibility of this paper may be good, the key resources (i.e., code and data) will be available upon publication as promised in the Abstract. However, the experiments should be improved.

**Strength And Weaknesses:**

Strengths:
1. The proposed task/problem is new. The constructed dataset seems to be useful for the community.
sound.
2. The proposed approach can outperform several constructed baseline models.

Weaknesses:
1. The usefulness of the proposed task should be clarified.
- What benefits can be brought by this task in clinical? And why?
- Otherwise, I may think that this work just applies the techniques/tasks in computer vision (i.e., NS-VQA and image difference captioning) to medical images without understanding the clinical significance of the problem being addressed.
- More importantly, the proposed task is very similar to the existing NS-VQA task [1], but the authors neither cite nor discuss this paper.
- Meanwhile, no mention is made of the type of abnormalities for which this approach/task is suitable. In fact, since a chest X-ray may show many pre-existing conditions which may be anatomical abnormalities (e.g. scoliosis) but not the target of the chief complaint for which the report was ordered, it is important to quantify the type of abnormalities.

2. The novelty of this work is limited.
- First, as mentioned in Weakness 1, the proposed medical image difference VQA task is very similar to the existing NS-VQA task [1], but the authors neither cite nor discuss this paper. Therefore, it's important to clarify what benefits can be brought by this task in clinical.
- The proposed approach and the arguments, i.e., "This is consistent with the radiologist’s diagnosis practice that compares the current image with the reference before concluding the report" in the Abstract and "When radiologists make a diagnosis, they usually compare the main one with a reference image to find their differences" in the Introduction, are very similar to the existing work [2].
- Figure 7(b) is directly borrowed from [3]. The authors should note it.

3. The motivation is unclear.
- The motivation for introducing the reference image is unclear.
- Does there are any relationships between the designed questions and the reference image?
- What are the contributions of the incorporated reference image?
- What are the reasons to adopt a reference image to answer the questions?
- Regarding the proposed approach, why does this work adopt the language model? What is the motivation?

4. The experiments should be improved.
- This paper only includes one baseline model for comparison. I strongly recommend the authors attempt to re-implement existing state-of-the-art methods in NS-VQA on the constructed dataset to show the effectiveness of the proposed approach.
- Could you provide the results of the approach on different types of question types/abnormalities?
-  I am also interested in knowing if the approach brings errors. And what type of errors does it bring? And why?

5. Some details of the pre-processing of the constructed dataset are missing.
- As the main contribution of this paper, it's necessary to provide more details about the dataset.
- Rule-based methods: What is the motivation for adopting rule-based methods? How to adopt the rule-based methods to construct the dataset?
- The MIMIC-CXR dataset includes two views of chest x-rays, how do you process them?
- Does the proposed approach and task could be applied to the lateral view image?
- Could you give some examples of the lateral view image?

6. (Minor) This paper seems to be written in a bit of a hurry, and there is a lot of scope for improving the presentation of the paper. I strongly recommend the author check and revise the text carefully.


References:

[1] Neural-Symbolic VQA: Disentangling Reasoning from Vision and Language Understanding. In NeurIPS, 2018.

[2] Contrastive Attention for Automatic Chest X-ray Report Generation. In ACL, 2021.

[3] When Radiology Report Generation Meets Knowledge Graph. In AAAI, 2020.

**Summary Of The Paper:**

This work proposes a medical image difference VQA problem and constructs a MIMIC-Diff-VQA dataset from the existing MIMIC-CXR dataset. To perform the task, this work proposes a knowledge-aware graph and a multi-relationship graph, where the former takes each anatomical structure as a node in the graph and compares the image differences in each anatomical structure, and the latter adopts the spatial relationship, semantic relationship, and implicit relationship to compute the image-difference graph representations. The experiments on the constructed MIMIC-Diff-VQA dataset show that the proposed approach

**Summary Of The Review:**

Overall, the paper has provided a new task and a new large-scale dataset, but the usefulness of the proposed task should be clarified. Meanwhile, the novelty of this paper should be discussed. Before the rebuttal, I tend to reject this paper.

---

> ### Author Response · Authors · 2022-11-19
> **Response to Reviewer FdAD (5/5)**
>
> ## 5. Some details of the pre-processing of the constructed dataset are missing.
>
> ### Q11. What is the motivation for adopting rule-based methods? How to adopt the rule-based methods to construct the dataset? more details?
>
> A11: The motivation of our work is to design an AI system with deep collaboration with radiologists and clinicians. Therefore, we incorporate label extraction rules to obtain knowledge from the radiologist and clinician explicitly. These rules also offer the window to let doctors understand and regularize the networks.
>
> Although current general vision language modeling provides some end2end approaches for extracting keywords to link image and text, these methods rely on pre-trained models on high-quality image and text dataset, and are generally black box and lack transparency/interpretability.  The amount of labeled medical image and text data is very limited due to the expert labeling cost and complex medical vocabularies, therefore, there are rarely available medical pre-trained vision and language models to enhance the end2end approaches in the medical text analysis domain. The rule-based method is more practical compared to the end2end deep learning approach in our problem.
>
> Due to page limits, we described the details about how the dataset is constructed in Appendix A.2. First, we extracted the abnormality/attribute keywords by frequency. After manually screening these keywords, the clinician further helped us to select the clinically important keywords. Next, we design the rule-based method to extract these selected keywords.
>
> ### Q12. how to process two views of chest x-rays?
>
> A12: As we mentioned in Appendix A.2, we currently only consider PA or PA views for comparing the main and reference images for simplicity. We will explore the lateral view images in future work.
>
> ## 6. Minors
>
> ### Q13. Writing improvement.
>
> A13: We have revised our presentation and improved the presentation in the revised paper.

---

> > ### Comment · Reviewer_FdAD · 2022-12-03
> > **A further question**
> >
> > Many thanks for the detailed rebuttal. I have increased my score from 3 to 5.
> > However, I have a further question. There are two important features or goals in supporting clinical decision-making:
> >
> > - Evidence. The prediction/generation should be displayed with means for the clinician to inspect and understand where the information comes from.
> > - Faithfulness. Measuring and understanding the faithfulness of the model output is important. I'm wondering if the model adopts the learned data distributions of the dataset (e.g., the connections between the question type and its corresponding frequent answers) to make predictions, instead of really understanding the medical input and medical knowledge. The "language priors" is a nutritious problem in general visual question answering.
> >
> > Therefore, can this approach achieve these two goals? Why? Meanwhile, I'm very interested in if the approach brings errors, and how you handle it. Because making inaccurate predictions in clinical practice is dangerous.

---

> > > ### Author Response · Authors · 2022-12-13
> > > **Response to Reviewer FdAD**
> > >
> > > Thank you for your enlightening and inspiring comments. The point you raised about evidence and faithfulness is crucial for the collaboration between doctors and AI. After further investigation, we discovered that our model has the ability to improve and enhance both Evidence and Faithfulness.
> > >
> > > ## Q1: Evidence.
> > > **A1:** 1, Our model can highlight the regions on the images that are associated with diseases, allowing doctors to quickly and easily inspect and verify their thoughts.
> > > 2, Additionally, our model allows doctors to further ask questions about the location of specific abnormalities, providing them with text location information to aid in their diagnoses. The medical diagnosis of disease generally undergoes a course-to-fine fashion, starting with presence diagnosis and progressing to precise localization, ultimately leading to a definitive diagnosis with more evidence. In this process, when the finer information is consistent with the course diagnosis, the doctors’ confidence in clinical decisions increases. For example, when given an image, a doctor will have an initial impression of what and where an abnormality is. The first question the doctor can ask the model is "What abnormalities are seen in this image?", and if the prediction is "atelectasis" and it matches the doctor's hypothesis, the doctor can then ask a second question: "Where is the atelectasis?", and if the answer is again consistent with the doctor's hypothesis, this will reinforce the doctor's confidence in the diagnosis. In this case, our model can act as a second examiner to assist the doctor in verifying their hypotheses during this confidence-building procedure.
> > >
> > >
> > >
> > > ## Q2: Faithfulness.
> > > **A2:**  Thank you for highlighting the distribution bias issue. As you pointed out, this is a common problem in many datasets. As [1] has demonstrated, using only image information to answer questions can yield good results, indicating the poor distribution of data in existing datasets. In response to this, we made efforts to ensure that our dataset was relatively balanced. The table below presents the statistics of our dataset, including the partial most frequent questions and the number of top-3 most frequent answers corresponding to each question.
> > >
> > > In order to verify the language prior problem, we performed another experiment by removing all images and only keeping the questions.  The resulting predictions were significantly worse than those obtained using the original images. Bleu-1 dropped from 0.63 to 0.51, Bleu-2 dropped from 0.54 to 0.33, Bleu-3 dropped from 0.48 to 0.18, and Bleu-4 dropped from 0.42 to 0.12. This shows that our method is not highly dependent on language priors and relies more on images.
> > > | Question                                                 | Answer1 # | Answer2 # | Answer3 # |
> > > |----------------------------------------------------------|-----------|-----------|-----------|
> > > | 'is there evidence of pleural effusion in   this image?' | 6559      | 4500      | -         |
> > > | 'is there pleural effusion?'                             | 6464      | 4477      | -         |
> > > | 'where is the pleural effusion?'                         | 3522      | 2408      | 2311      |
> > > | 'where in the image is the pleural   effusion located?'  | 3458      | 2425      | 2277      |
> > > | 'what abnormalities are seen in this   image?'           | 3811      | 2569      | 2539      |
> > > | 'what type is the lung opacity?'                         | 3469      | 3417      | 1921      |
> > >
> > > [1]Gong, H., Huang, R., Chen, G., & Li, G. (2021). Sysu-hcp at vqa-med 2021: A data-centric model with efficient training methodology for medical visual question answering. Proceedings http://ceur-ws. org ISSN, 1613, 0073.
> > >
> > > ## Q3: errors
> > > **A3:** Representative errors can be summarized into three types: 1, confusion between different presentation aspects of the same abnormality, such as atelectasis and lung opacity being mistaken for each other. 2, different names for the same type of abnormality, such as enlargement of the cardiac silhouette being misclassified as cardiomegaly. 3, the pre-trained backbone (Faster-RCNN) used for extracting image features may provide inaccurate features and lead to incorrect predictions, such as lung opacity being wrongly recognized for pleural effusion.

---

> ### Author Response · Authors · 2022-11-19
> **Response to Reviewer FdAD (4/5)**
>
> ## 4. The experiments should be improved.
>
> ### Q8. compare to NS-VQA or other baselines on the constructed dataset?
>
> A8: As mentioned in [Q2](#Q2:-difference-to-NS-VQA.), NS-VQA is not designed for image difference questions and answers. It was applied to a simple synthetic dataset and was unsuitable for our challenging real-world dataset. However, we have compared our method with the state-of-the-art method on the image difference captioning task, as shown below.
>
>
> > Yao, Linli, Weiying Wang, and Qin Jin. "Image difference captioning with pre-training and contrastive learning." arXiv preprint arXiv:2202.04298 1.4 (2022).
>
> ##### Comparison with baseline Yao22:
>
> We have added the most recent general image-difference caption model (IDC)(Yao 22) on the challenging birds-to-words dataset  (complicated background and large differences between compared image pairs) to our experiments. We modified it for the image-difference VQA problem by feeding the image pairs of different question types only,  and compared our method to it.  Our method demonstrated significant improvement compared to *Image Difference Captioning with Pre-training and Contrastive Learning* since we extracted image difference features and generated answers by leveraging clinical knowledge such as anatomical structure and knowledge graphs.
>
>
> We show that their performance on our constructed dataset is: Bleu1: 0.525, Bleu2: 0.464, Bleu3: 0.405, Bleu4: 0.354, which is lower than ours: Bleu1: 0.641, Bleu2: 0.564, Bleu3: 0.500, Bleu4: 0.441.  Moreover, in terms of the CIDEr, a metric that measures the similarity between sentences, our method achieved 0.654, whereas IDC only reached 0.318. Meanwhile, by looking into the output examples of IDC, their outputs are mostly focusing on the unimportant part but missing the clinically important keywords.
>
> For example, for the question “what has changed compared to the reference image?” for the image pair between study id 52246202 and 58366233, the ground truth answer is “the main image is missing the finding of atelectasis than the reference image.  “ with clinical keywords: atelectasis.
>
> The answer from IDC is   “costophrenic the main image has additional findings of lung opacity , and cardiomegaly than the reference image . the main image is missing the findings of cardiomegaly , and than the the reference image . ” The IDC method missed the clinical important keywords: atelectasis. they also generated other unrelated keywords, such as cardiomegaly, lung opacity.
>
> And the answer of our method is: “the main image is missing the finding of atelectasis than the reference image.  ”, which perfectly matches the ground truth answer. Our answer captured the clinical important keywords:atelectasis.
>
> ### Q9. Results on different types of questions/abnormalities?
>
> A: We listed the results of different question types in the following table. The table has been added to the revised paper in Table 11.
>
> | question type | Bleu 1 | Bleu 2 | Bleu 3 | Bleu 4 |
> |---------------|--------|--------|--------|--------|
> | abnormality   | 0.482  | 0.333  | 0.197  | 0.109  |
> | presence      | 0.801  | -      | -      | -      |
> | view          | 0.948  | 0.941  | -      | -      |
> | location      | 0.525  | 0.364  | 0.210  | 0.144  |
> | level         | 0.496  | 0.101  | 0.068  | -      |
> | difference    | 0.641  | 0.564  | 0.500  | 0.441  |
>
> As shown in the table, Bleu 3 and Bleu 4 tend to have low scores. This is because the answers to most of the questions are short, except for the "difference" questions. For abnormality questions, 72% of the answers have less than or equal to 2 words; for location questions, 79% have less than or equal to 2 words; 93% of level questions have one-word answers.
>
> ### Q10. what type of errors does it bring? And why?
>
> A10: It is worth noting that we are working on real-world challenging datasets and the abnormalities/diseases are highly unbalanced in our datasets. The detection and classification results of some abnormalities are much lower than others due to the small number of samples in training data.  However, these challenges in our dataset also provide great opportunities for researchers in few-shot learning, zero-shot learning, and unbalanced dataset learning. We believe our dataset can be used by researchers in the general machine learning modeling domain to evaluate the benchmarks of the few, zero-shot learning, and unbalanced dataset learning.

---

> ### Author Response · Authors · 2022-11-19
> **Response to Reviewer FdAD (3/5)**
>
> ## 2. The novelty of this work is limited.
>
> ### Q4: Difference to *Contrastive Attention for Automatic Chest X-ray Report Generation.  ACL, 2021.*
>
> A4: The focus of our work is different from the ACL 2021 paper. We focused on comparing the past visiting and current visiting images from the same patient to track the subtle changes that happened between the two visits. Our method is clinically driven and aims at helping the radiologist validate the hypothesis of what has changed after the intervention for each patient.
>
> The ACL 2021 paper focused on extracting abnormality-related image features by constructing a pool of normal chest x-ray images and using contrastive learning to distill the contrastive features between abnormal and normal images to improve the report generation performance.  We cite and discuss this paper in Related Work in Appendix A.1 marked in blue.
>
> ### Q5: Fig 7(a) is from [3]?
>
> A5: We built our knowledge graph upon the success of previous work [3], which leveraged the knowledge graph to improve chest X-ray classification. However, we modified the knowledge graph from [3] by including more abnormalities, such as aortic enlargement, enlarged PA, consolidation, ILD, infiltration, pulmonary fibrosis, nodule/mass, lung cyst, lung cavity, mediastinal shift, and calcification. We also removed the elements that are not highly related to our task, such as “spine”, “foreign object”, “airspace”, and “other finding”.
>
> We have modified the figure of the knowledge graph in the revised paper(Figure 8(a)) by highlighting the newly added abnormalities using red color.
>
>
> ## 3. The motivation is unclear.
>
> ### Q6: The motivation of the reference image?
>
> A: As mentioned in [Q1](#1.-The-usefulness-of-the-proposed-task-should-be-clarified.), the reference image serves as the past visiting image so that we can compare the current image to find out what has changed since the intervention of the last visit.
>
> ### Q7: motivation of language model?
>
> A: With the rapid development of the pre-trained language model, the current state-of-art general VQA model mostly uses language models. We follow these state-of-art works to use language models.  In addition, we have a substantial and diverse answer candidate pool (51040 answers) in our dataset due to the complicated disease relationships. Training a simple one-hot encoding classification model for these complicated questions and answers is not practical. Language models can capture the semantic relationship between questions and complicated answers to generate semantic meaningful answers.

---

> ### Author Response · Authors · 2022-11-19
> **Response to Reviewer FdAD (2/5)**
>
> ### Q2: difference to NS-VQA.
>
> A2: NS-VQA extracts regions of interest (ROIs) with predicted semantic labels and generates scene graphs based on the semantic labels using Mask-RCNN. However, NS-VQA focused on leveraging pre-designed python logical programs to process different questions and interpret(calculate) the answers. NS-VQA’s answer generation greatly relies on the quality of the object segmentation and labeling by pre-trained Mask-RCNN. Since NS-VQA only evaluated the performance on a simple dataset: CLVER, where all pictures have a single color background, each object has a fixed number of labels and the same label types. Thus, training Mask-RCNN to detect different objects on this dataset is easy to obtain an ideal performance.
>
> We do not think NS-VQA is suitable for a challenging real-world image dataset like our dataset, where the image background is complicated,  and each region of interest in the image has a different number of potential labels and different label types. Lastly and most importantly, NS-VQA did not consider the image-difference feature extraction and image-difference question answer problem at all.
>
> We focused on a novel image-difference VQA problem which is clinically significant since it follows the way radiologists diagnose disease and check the results of after-interventions. To solve this problem, we designed an anatomical structure-aware image difference feature extraction model which embedded the knowledge of how radiologists read and compare the chest X-ray images, which is never considered in the general image difference caption or general visual question answer methods.
>
> In addition, we used both the pre-trained feature for anatomical region and disease region detection, which allows our graph-based model to process more comprehensive features to generate the final answers thoroughly. Different anatomical structure features may provide different levels of context information for answering specific questions such as “where in the image is the pleural effusion located ?”
>
> We compared to another strong baseline *Yao22.* As shown in [Q8](#Q8.-compare-to-NS-VQA-or-other-baselines-on-the-constructed-dataset?).
>
>
> ### Q3: no mention is made of the type of abnormalities for which this approach/task is suitable.
>
> A3: We have listed each extractable abnormality in Table 6 in the Appendix due to the page limits.
>
> | Applicable abnormalities              |
> |---------------------------------------|
> | pleural effusion                      |
> | atelectasis                           |
> | cardiomegaly                          |
> | enlargement of the cardiac silhouette |
> | edema                                 |
> | hernia                                |
> | vascular congestion                   |
> | hilar congestion                      |
> | pneumothorax                          |
> | heart failure                         |
> | lung opacity                          |
> | pneumonia                             |
> | tortuosity of the descending aorta    |
> | scoliosis                             |
> | gastric distention                    |
> | hypoxemia                             |
> | hypertensive heart disease            |
> | hematoma                              |
> | tortuosity of the thoracic aorta      |
> | contusion                             |
> | emphysema                             |
> | granuloma                             |
> | calcification                         |
> | pleural thickening                    |
> | thymoma                               |
> | blunting of the costophrenic angle    |
> | consolidation                         |
> | fracture                              |
> | pneumomediastinum                     |
> | air collection                        |

---

> ### Author Response · Authors · 2022-11-19
> **Response to Reviewer FdAD (1/5)**
>
> Thank you for your careful comments! We have addressed your kind concerns and advice in our revised paper.
>
> ## 1. The usefulness of the proposed task should be clarified.
> ### Q1: What benefits can be brought by this task in clinical? And why?
>
> A1: The two main benefits our task can bring to the clinical application are 1, our task is consistent with clinical practice. 2, we provide the tools for doctors to verify their clinical hypotheses.
>
> Existing medical image analysis and VQA focused on applying existing general vision methods to medical imaging with a single image as input. However, actual diagnosis by radiologists involves comparing current and previous images of the same patients to check the disease's progress. Actual clinical practice follows a patient treatment process (assessment - diagnosis - intervention - evaluation).
>
> A baseline medical image is used as an assessment tool to diagnose a clinical problem, usually followed by therapeutic intervention. Then, another follow-up medical image is retaken to evaluate the effectiveness of the intervention in comparison with the past baseline.  In this framework, every medical image has its purpose of clarifying the doctor's clinical hypothesis depending on the unique clinical course (e.g., whether the pneumothorax is mitigated after therapeutic intervention).
>
> However, existing methods fail to provide a straightforward answer to the clinical hypothesis because of the lack of focus on the clinical course that can be considered by comparing the past and present images. In contrast, our system responds directly to the information the doctor wants by comparing the current medical image (main) to a past visit medical image (reference). This allows us to build a diagnostic support system that realizes the inherently interactive nature of radiology reports in clinical practice.
>
> In addition,  radiologists always refer to examining anatomical structures to read Chest X-Ray images and comparing the changes between current visit and past visit chest x-ray images. To follow how radiologists interpret medical images, we designed an anatomical structure-aware image difference graph to extract the clinically significant changing features between two comparing images in different anatomical regions (such as left upper lung, right middle lungs, etc.).
>
> At last, we designed a multi-relationship graph that leveraged both spatial and semantic relationships to answer different medical image difference questions. The semantic relationship is constructed based on extracted prior knowledge from clinical literature with disease-to-anatomical structure and disease-to-disease relationship.  The multi-relationship graph can interpret how the answer is generated for different questions.
>
> To this end, our proposed medical image difference VQA framework can be deeply integrated with the clinical diagnostic process, enhancing the collaboration between radiologists and AI, and relieving radiologists from the stress of repetitive diagnoses work by providing preliminary AI-generated answers. The AI-generated answers can also be used by patients as a reference to answer their most important concerns when the radiologist is not available.

---

### Official Review · Reviewer_eNEk · 2022-11-08

**Confidence:** 3
**Correctness:** 3
**Technical Novelty And Significance:** 2
**Empirical Novelty And Significance:** 2
**Recommendation:** 5

**Clarity, Quality, Novelty And Reproducibility:**

Clarity:
- Some errors in grammar (e.g. sentences beginning "while the current state-of-the-art ...." is an unfinished sentence).
- Future versions should place citations inside brackets. Currently it's a little tough to read.
- I found the model architecture and decsription difficult to follow.
- In the section `Multi-Relationship Graph Module`, a number of symbols in the graph definition are left undefined, e.g. $\varepsilon_{sp}$, and the symbols in the definition for vertices. (They can be inferred from context, but they should still be defined in the text).
- CIDEr metric not explained.
- The difference between contributions 2 and 3 is a bit unclear.
- [minor] "MCCFormer" sometimes written as "MMCFormer"
- [minor] opening quote brackets (") are the wrong way around
- [minor] spelling errors: "relaiton", "perfomrd",
- [minor] should bold "ours" results in Table 3.

Novelty:
- The dataset novelty is good.
- Unsure about novelty in the knowledge-graph model component

Reproducibility:
- Unfortunately I cannot yet access the code or dataset, so I cannot rate reproducibility as a strength.

**Details Of Ethics Concerns:**

The dataset is built on top of existing public datasets, so this should not raise new issues.

**Strength And Weaknesses:**

Strengths:
- The dataset is useful and well-motivated. The authors argue that radiologists really do evaluate images by considering differences in pairs of images from the same patient.
- Furthermore, the dataset is real with clear future applications. Hopefully it can be used as a challenging benchmark beyond existing toy examples in similar tasks like 'image captioning'.
- The method far outperforms the baseline.

Weaknesses:
- Concern about dataset correctness: Fig.1 and introduction paragraph 1 argue that prior VQA datasets in medical images have significant errors in their labels due to issues with their text mining strategies (for example, issues with rule-based systems). But paragraph 3 claims that similar approaches are used to generate the new dataset. It is good that some human verification was done (as explained in the appendix), but only 300 randomly chosen pairs were checked out of from 700,821. Also, the dataset is not yet released, so reviewers cannot check. This contribution would be stronger by a more thorough verification.
- For the method, the ablations in Table 2 suggest marginal gains for the spatial and semantic graphs, so the model could be simplified.
- MCCFormer (Qiu 21) is the only image difference captioning baseline, and it does badly on the proposed task. The authors argued this is because they work patch-wise and so cannot handle mis-aligned pairs (which isn't a problem for datasets like CLEVER-change). But the related work discuss "birds-to-words" dataset which has misaligned pairs, and it mention several works that perform well on it: (Yao 22) and (Yan 21). Therefore these works may be appropriate baselines to include.



**Summary Of The Paper:**

First, the paper introduces a dataset for "difference-aware medical visual question answering". It has chest x-ray pairs from the same patient, with associated questions and answer labels for a VQA task. It is scraped from the existing MIMIC dataset. Though this is similar to "image captioning" task, the authors argue that medical imaging comparisons have distinct challenges, so this should be considered a distinct task.

Second, the paper proposes a model for solving this proposed task. This model uses a detector to identify anatomical regions used for feature extraction, which are then used in a knowledge graph model.


**Summary Of The Review:**

The dataset is a useful contribution, and I hope it is used more widely, especially by general VQA researchers. This contribution would be stronger if it is verified more thoroughly (as discussed above). I have a concerns regarding baselines and clarity (see above).

---

> ### Author Response · Authors · 2022-11-19
> **Response to Reviewer eNEk**
>
> Thank you for your insightful comments! We have addressed your concerns in our revised paper.
>
> ### Q1. why blame rule-based label extraction and still use rule-based approach in VQA.
>
> A1: Current rule-based label extraction method only focused on a small set of disease-related labels considering minimum negations, and no complicated disease pathologies, level, and location information. Therefore, their extracted labels lack clinical important information and have significant errors caused by negations.  Also, we followed an Extract-Check-Fix cycle to customize the rule set for MIMIC to consider all important information and minimize errors when constructing our dataset, and performed extensive manual verification to ensure the dataset's quality.
>
> We propose a medical VQA framework, which directly answers the clinically important questions related to disease type, locations, level, and changes. Our VQA dataset is generated by setting the rules based on the radiologist's required information. We are the first work focusing on the medical image difference VQA problem with radiologist knowledge-guided rules for important clinical label extraction and visual question-answer dataset construction.  Encoding the clinical important rules to train the model can also help to interpret why and how the diagnosis is made.
>
> ### Q2. more human verifications.
>
> A2: We have further validated another 1700 examples, and the accuracy reached  97.4% by our rule-based automatic label extraction method. The new result has been updated in the revised paper.
>
>
> ### Q3: spatial and semantic graphs show marginal gains, why need them?
>
> A3: Although our spatial and semantic graphs get marginal gains on the Bleu scores, it helps to improve the VQA interpretation.  As shown in Appendix A.8,  we visualized the interpretation of the implicit graph versus adding semantic and spatial graphs, which demonstrated that semantic and spatial information help to improve the interpretability of this work.
>
>
> ### Q4: compared to strong baseline: Yao 22
>
> > Yao, Linli, Weiying Wang, and Qin Jin. "Image difference captioning with pre-training and contrastive learning." arXiv preprint arXiv:2202.04298 1.4 (2022).
>
> A4: We have added the most recent general image-difference caption model (IDC)(Yao 22) on the challenging birds-to-words dataset  (complicated background and large differences between compared image pairs) to our experiments. We modified it for the image-difference VQA problem by feeding the image pairs of different question types only,  and compared our method to it.
>
> We show that their performance on our constructed dataset is: Bleu1: 0.525, Bleu2: 0.464, Bleu3: 0.405, Bleu4: 0.354, which is lower than ours: Bleu1: 0.641, Bleu2: 0.564, Bleu3: 0.500, Bleu4: 0.441.  Moreover, in terms of the CIDEr, a metric that measures the similarity between sentences, our method achieved 0.654, whereas IDC only reached 0.318. Meanwhile, by looking into the output examples of IDC, their outputs are mostly focusing on the unimportant part but missing the clinically important keywords.
>
> For example, for the question “what has changed compared to the reference image?” for the image pair between study id 52246202 and 58366233, the ground truth answer is “the main image is missing the finding of atelectasis than the reference image.  “ with clinical keywords: atelectasis.
>
> The answer from IDC is   “costophrenic the main image has additional findings of lung opacity , and cardiomegaly than the reference image . the main image is missing the findings of cardiomegaly , and than the the reference image . ” The IDC method missed the clinical important keywords: atelectasis. they also generated other unrelated keywords, such as cardiomegaly, lung opacity.
>
> And the answer of our method is: “the main image is missing the finding of atelectasis than the reference image.  ”, which perfectly matches the ground truth answer. Our answer captured the clinical important keywords:atelectasis.
>
> ### Q5: grammar, typos, reference, and other minors.
>
> A5: We have fixed these issues in the revised paper.

---

### Author Response · Authors · 2022-12-13
**Thanking Reviewers for Their Valuable Insights**

We are pleased to discover that the reviewers consider our proposed task/problem to be both new (Reviewer FdAD) and well-motivated with clear potential applications (Reviewer eNEk). They also appreciate the value of our large-scale dataset in advancing multi-modal models and mimicking the work of radiologists when comparing studies(Reviewer NUDS). Additionally, they have confirmed the novelty of our knowledge graph representation (Reviewer NUDS).

As the reviewers have pointed out, there is indeed a gap in the existing medical VQA datasets, and we are delighted to have contributed to filling this gap with our proposed dataset. We hope to have the opportunity to discuss this problem further with you at the upcoming ICLR conference.

Thank you again for your valuable insights and suggestions. Your feedback is greatly appreciated and has been incredibly useful in improving our method.

---

### Decision · Program_Chairs · 2023-01-20

**Decision:**

Reject

**Justification For Why Not Higher Score:**

The modeling and evaluation concerns are too significant for acceptance.


**Justification For Why Not Lower Score:**

N/A

**Metareview: Summary, Strengths And Weaknesses:**

This paper received borderline ratings overall, with some divergence across reviewers, and was therefore additionally discussed in a virtual meeting with reviewers. Reviewers overall appreciated the value of the new dataset. However, multiple reviewers expressed concerns with the proposed model and experiments, even after the author rebuttal. In particular, although the model was proposed to address medical VQA, it was not sufficiently convincing that the main components of the model that contributed to performance (as presented in ablation studies) were targeted to address medical image-specific challenges, and that it makes sense to distinguish comparison of this model and evaluation from general VQA models and evaluation. Additionally, it was not clear that the work has sufficiently compared against strong VQA baselines that could also be adapted to the medical VQA task, including several references suggested by the reviewers. The author rebuttal partially but did not completely satisfy reviewer concerns about this. Overall, after the reviewer meeting, two reviewers chose to keep their recommendation as reject, while the third reviewer with the highest score clarified that their score was primarily based on evaluation of the dataset and they did not oppose rejection due to the modeling and evaluation concerns. I agree with the majority reviewer opinion that the weaknesses of this paper outweigh the strengths in the current form and do not recommend acceptance at this time.

**Summary Of Ac-Reviewer Meeting:**

Reviewers primarily had modeling and evaluation concerns, that were partially but not satisfactorily addressed by the authors. These points weighed significantly in the decision, while the reviewer with the highest score clarified that their score was primarily based on evaluation of the dataset and they did not oppose rejection due to the modeling and evaluation concerns. I do not think the dataset alone is sufficient to justify acceptance of the paper.